# FACTSCORE: Fine-grained Atomic Evaluation of Factual Precision in Long Form Text Generation

**Sewon Min**[†1] **Kalpesh Krishna**[†2] **Xinxi Lyu**[1] **Mike Lewis**[4] **Wen-tau Yih**[4]
**Pang Wei Koh**[1] **Mohit Iyyer**[2] **Luke Zettlemoyer**[1,4] **Hannaneh Hajishirzi**[1,3]

[1]University of Washington [2]University of Massachusetts Amherst
[3]Allen Institute for AI [4]Meta AI
{sewon,alrope,pangwei,lsz,hannaneh}@cs.washington.edu
{kalpesh,miyyer}@cs.umass.edu {mikelewis,scottyih}@meta.com

## Abstract

Evaluating the factuality of long-form text generated by large language models (LMs) is non-trivial because (1) generations often contain a mixture of supported and unsupported pieces of information, making binary judgments of quality inadequate, and (2) human evaluation is time-consuming and costly. In this paper, we introduce **FACTSCORE**, a new evaluation that breaks a generation into a series of atomic facts and computes the percentage of atomic facts supported by a reliable knowledge source. We conduct an extensive human evaluation to obtain FACTSCOREs of people biographies generated by several state-of-the-art commercial LMs—InstructGPT, ChatGPT, and the retrieval-augmented PerplexityAI—and report new analysis demonstrating the need for such a fine-grained score (e.g., ChatGPT only achieves 58%). Since human evaluation is costly, we also introduce an automated model that estimates FACTSCORE using retrieval and a strong language model, with less than a 2% error rate. Finally, we use this automated metric to evaluate 6,500 generations from a new set of 13 recent LMs that would have cost $26K if evaluated by humans, with various findings: GPT-4 and ChatGPT are more factual than public models, and Vicuna and Alpaca are some of the best public models. FACTSCORE is available for public use via `pip install factscore`.[1]

## 1 Introduction

Long-form text generated by large language models (LMs) has widely been used (Brown et al., 2020; Ouyang et al., 2022); nonetheless, evaluating their *factual precision*—whether each piece of information conveyed in a generation is factually accurate—remains challenging for two reasons. First, a generation consists of a large number of pieces of infor-

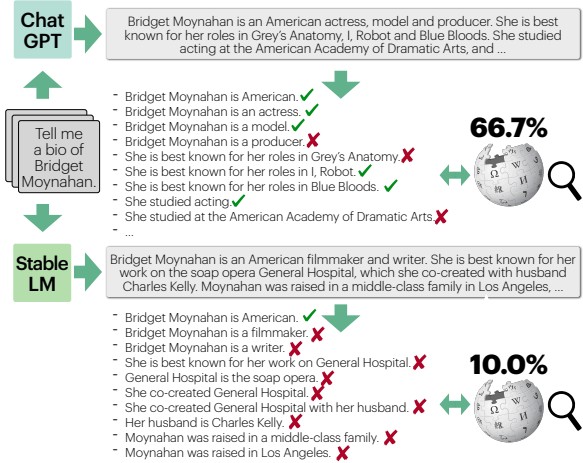

Figure 1: An overview of FACTSCORE, a fraction of *atomic facts* (pieces of information) supported by a given knowledge source. FACTSCORE allows a more fine-grained evaluation of factual precision, e.g., in the figure, the top model gets a score of 66.7% and the bottom model gets 10.0%, whereas prior work would assign 0.0 to both. FACTSCORE can either be based on human evaluation, or be automated, which allows evaluation of a large set of LMs with no human efforts.

mation that are a mixture of true or false,[2] making a binary judgment inadequate (Pagnoni et al., 2021). Second, validating every piece of information is time-consuming and costly.

In this paper, we introduce **FACTSCORE** (**F**actual precision in **A**tomi**c**ity **Score**), a new evaluation of an LM that represents *the percentage of atomic facts (pieces of information) supported by a given knowledge source*. Computing FACTSCORE involves (1) breaking a generation into a series of atomic facts—short statements that each contain one piece of information (Nenkova and Passonneau, 2004; Shapira et al., 2019; Zhang and Bansal, 2021; Liu et al., 2022), and (2) assigning a binary label

---

[†]Core contributors.
[1]Source code and guidelines are available at https://github.com/shmsw25/FActScore.

[2]Even a single sentence consists of multiple pieces of information (e.g., 4.4 per sentence in ChatGPT, 40% of which are a mixture of supported and unsupported information).

to each atomic fact, allowing a fine-grained evaluation of factual precision. We evaluate FACTSCORE on the task of generating people biographies because generations consist of verifiable statements rather than debatable or subjective ones, and the scope is broad (i.e., covering diverse nationalities, professions, and levels of rarity).

We perform extensive human annotations to obtain FACTSCOREs of three state-of-the-art, commercially available LMs: InstructGPT (Ouyang et al., 2022), ChatGPT (OpenAI, 2022), and search-augmented PerplexityAI.[3] Our results indicate that commercially available LMs are riddled with errors, having FACTSCOREs of 42%, 58% and 71%, respectively. Their FACTSCOREs significantly drop as the rarity of the entities increases, e.g., $80\% \rightarrow 16\%$ for ChatGPT.

Since human evaluation is costly, we next introduce an automatic evaluation of FACTSCORE through a model that estimates a FACTSCORE for a given LM. Our estimator decomposes generations into atomic facts and validates each based on a given knowledge source, leveraging retrieval from the given knowledge source and strong language models. Our estimator closely approximates FACTSCORE with an error rate of $< 2\%$ and can be applied to a range of *new* LMs at scale with no human effort. Our case study evaluates 6,500 generations from 13 LMs that could have cost $26K, with various findings: GPT-4 (OpenAI, 2023) and ChatGPT are far less factual than humans but are much better than public models, and there is a large variance between public models, with Vicuna (Chiang et al., 2023) and Alpaca (Taori et al., 2023) being some of the best.

In summary, our contributions are as follows.

1. We introduce FACTSCORE, a new evaluation of factual precision of LMs by breaking their generations into atomic facts and validating each against a given knowledge source. Human evaluation reveals that the state-of-the-art LMs with and without search have low FACTSCOREs.

2. We introduce a model that approximates FACTSCORE with an error rate of $< 2\%$, allowing evaluation of a large set of new LMs without manual human efforts.

3. We open-sourced FACTSCORE and the annotated data for public use, available via `pip install factscore`. We suggest future work to extend FACTSCORE for a broader set of generations (e.g., open-ended generation) and to further improve the estimator.

## 2 Related Work

**Factual precision in text generation.** Factual precision in text generation has been an active area of research in NLP. Most prior work studies factual precision of models supervised for a specific problem such as dialogue (Shuster et al., 2021), or focuses on question answering with short answers (Kadavath et al., 2022; Kandpal et al., 2022; Mallen et al., 2023; Nori et al., 2023).

More recent work has studied factual precision of text generation beyond short answers. Lee et al. (2022) evaluates the factual precision with proxy metrics, e.g., whether named entities in a generation appear in an article of the topic. A series of concurrent work verifies the precision of the citations (attributions) provided by the model (Gao et al., 2022; Liu et al., 2023a; Yue et al., 2023; Gao et al., 2023). A concurrent work by Manakul et al. (2023) automates the identification of factual errors in LM generations without using any knowledge source; we use their method as a baseline estimator in Section 4. In contrast, our work (1) considers much longer text generation[4] from a variety of state-of-the-art LMs with and without search, (2) provides their fine-grained evaluation both by human experts and through an automated evaluator that closely approaches humans, and (3) applies it to a large set of LMs at scale.

**Fact Verification.** Our work is closely related to prior work on fact verification (Thorne et al., 2018; Wadden et al., 2020) where claim sentences are automatically checked against a large knowledge source like Wikipedia or scientific literature. Most literature assumes a single, atomic claim, sometimes modeled with surrounding context (Nakov et al., 2018; Mihaylova et al., 2019; Shaar et al., 2022). There also has been work that verifies a longer sentence or text through decomposition to atomic facts (Fan et al., 2020; Wright et al., 2022; Chen et al., 2022; Kamoi et al., 2023) from which we take inspiration. The primary difference between fact verification literature and our work is that we focus on long-form *model-generated* text rather than sentence-level human-written claims.

---

[3]perplexity.ai

[4]Consisting of 110–151 words (Table 1), in contrast to 18–29 in Gao et al. (2022) and 65 in Liu et al. (2023a).

**Model-based Evaluation.** Prior work has used learned models to define automated evaluation scores (Zhang et al., 2020; Liu et al., 2023b). This includes model-based evaluation in summarization that considers the consistency between a summary and a source document using QA or NLI (Kryscinski et al., 2020; Wang et al., 2020; Fabbri et al., 2022; Deutsch et al., 2021; Laban et al., 2022). We take inspiration from this work, and evaluate factual precision of LM generations by considering whether pieces of information are supported by a large text corpus.

# 3 FACTSCORE: Evaluating Factual Precision of Long-form Text Generation

We introduce FACTSCORE, a new evaluation of an LM that considers the factual precision of atomic facts generated by the LM. We perform human evaluations to calculate FACTSCOREs of the state-of-the-art LMs (Section 3.3) and discuss results (Section 3.4). FACTSCORE allows rigorous and fine-grained evaluation of factual precision, but is time-consuming and costly, motivating automatic evaluation in Section 4.

## 3.1 Definition

FACTSCORE is based on two key ideas.

**Key idea 1: Atomic fact as a unit.** Long-form text consists of many pieces of information that can each be either true or false. Prior work has explored using a sentence as a unit; however, even a single sentence is a mix of supported and unsupported facts, e.g., in 40% of the cases with ChatGPT. Previous and concurrent work either (1) defines an additional label of `partial support` (Manakul et al., 2023; Liu et al., 2023a) whose definition may be subjective and can lead to low agreement, or (2) takes the strictest definition of `support` that requires every piece of information to be supported (Rashkin et al., 2021; Gao et al., 2022), which ignores the partial support cases, e.g., assigning 0.0 to both generations in Figure 1 even though the first generation is considerably more accurate than the second.

In this paper, we define an atomic fact as a short sentence conveying one piece of information (examples in Figure 1), similar to summarization content units (Nenkova and Passonneau, 2004). An atomic fact is a more fundamental unit than a sentence for a piece of information and provides a more fine-grained evaluation, e.g., in Figure 1, rat-

ing the first generation higher than the second.

**Key Idea 2: Factual precision as a function of a given knowledge source.** Prior work often considers factual precision as a single global truth (Manakul et al., 2023). In contrast, we adopt a perspective that the truthfulness of a statement should depend on a particular knowledge source that end users consider to be trustworthy and reliable. Therefore, instead of whether an atomic fact is globally true or false, we consider whether it is *supported* by a given source of knowledge. This has been used in the fact verification literature (Wadden et al., 2022) where conflict of information between different sources is relatively common.

**Definition.** Let $\mathcal{M}$ be a language model to be evaluated, $\mathcal{X}$ be a set of prompts, and $\mathcal{C}$ be a knowledge source. Consider a response $y = \mathcal{M}_x$ for $x \in \mathcal{X}$ and $\mathcal{A}_y$, a list of atomic facts in $y$. A FACTSCORE of $\mathcal{M}$ is defined as follows.

$$f(y) = \frac{1}{|\mathcal{A}_y|} \sum_{a \in \mathcal{A}_y} \mathbb{I}[a \text{ is supported by } \mathcal{C}],$$

FACTSCORE$(\mathcal{M}) = \mathbb{E}_{x \in \mathcal{X}}[f(\mathcal{M}_x) | \mathcal{M}_x \text{ responds}].$

$\mathcal{M}_x$ *responds* means $\mathcal{M}$ did not abstain from responding to the prompt $x$. This definition assumes the following:

1. Whether or not an atomic fact is supported by $\mathcal{C}$ is undebatable.
2. Every atomic fact in $A_y$ has an equal weight of importance, following Krishna et al. (2023).
3. Pieces of information in $\mathcal{C}$ do not conflict or overlap with each other.

In the rest of the paper, we propose to use people biographies as $\mathcal{X}$ and Wikipedia as $\mathcal{C}$ because they satisfy these assumptions to a reasonable degree (Section 3.3). We discuss in which cases these assumptions hold or may not hold in more detail in the Limitation section.

FACTSCORE considers *precision* but not *recall*, e.g., a model that abstains from answering too often or generates text with fewer facts may have a higher FACTSCORE, even if these are not desired. We leave the evaluation of factual recall for future work (more discussion in the Limitation section).

## 3.2 Studied LMs

We evaluate three LMs (referred to as LM$_{\text{SUBJ}}$, an LM as a subject): (1) **InstructGPT** (`text-davinci-003`, updated from Ouyang et al.

(2022)), (2) **ChatGPT** (OpenAI, 2022), and (3) **PerplexityAI**,[3] which incorporates a search engine with a language model.

## 3.3 Data

We perform human evaluation of factual precision based on our definition. We prompt the $LM_{SUBJ}$ to generate *people biographies* and evaluate them against Wikipedia for the following reasons.

- Biographies are objective (not subjective or debatable) and contain specific (not vague) information, satisfying Assumption 1 in Section 3.1.
- Biographies allow evaluation across diverse nationalities, professions, and levels of rarities.
- Wikipedia offers reasonable coverage of information about people and is reasonably self-consistent,[5] satisfying Assumption 3.

**Data collection.** We carefully design an annotation pipeline to assign a factual precision to a long-form generation through the following steps.

**Step 0: Sampling people entities.** We sample 183 people entities from Wikidata who have corresponding Wikipedia pages. We sample entities to annotate from a uniform distribution over categories defined in Appendix A.1.

**Step 1: Obtaining generations.** We feed a prompt "Tell me a bio of <entity>" to the $LM_{SUBJ}$ and take a generation as it is. We implement rules to identify generations that abstain from answering and filter them out.

**Step 2: Atomic facts generation.** Human annotators break a generation into a series of atomic facts. To save annotation time, we provide atomic facts broken down by InstructGPT which human annotators can take and revise. Details in Appendix A.2.

**Step 3: Labeling factual precision & editing.** We ask another set of human annotators to assign each atomic fact one of three labels. If the atomic fact is clearly not related to the prompt, and thus should be removed from the bio without a validation step, they assign Irrelevant. If the fact is relevant, they validate the fact based on the English Wikipedia, and label either Supported or Not-supported.

We recruit freelancers through Upwork and pay 15–25 USD per hour. Annotation requires extensive effort and time, leading to the cost of $4 per generation. We assign two freelancers for the 10%

[5]See Appendix A.5 for a related analysis.

|  | InstGPT | ChatGPT | PPLAI |
|---|---|---|---|
| Use search | ✗ | ✗ | ✓ |
| % responding | 99.5 | 85.8 | 90.7 |
| # tokens / response | 110.6 | 154.5 | 151.0 |
| # sentences / response | 6.2 | 7.9 | 9.8 |
| # facts / response | 26.3 | 34.7 | 40.8 |
| *Statistics of the labels* | | | |
| Supported | 42.3 | 50.0 | 64.9 |
| Not-supported | 43.2 | 27.5 | 11.1 |
| Irrelevant | 14.0 | 8.3 | 14.8 |
| Abstains from answering | 0.5 | 14.2 | 9.3 |
| **FACTSCORE** | **42.5** | **58.3** | **71.5** |

Table 1: Statistics of the data and FACTSCORE results. InstGPT and PPLAI respectively refer to InstructGPT and PerplexityAI. *% responding* indicates % of generations that do not abstain from responding. *# tokens* is based on white space.

of the data and calculate the agreement rate: 96%, 90% and 88% for InstructGPT, ChatGPT and PerplexityAI, respectively. More details are provided in Appendix A.3.

## 3.4 Results

Statistics of the data and results are reported in Table 1.

**All $LM_{SUBJ}$'s struggle with factual precision errors.** InstructGPT and ChatGPT achieve FACTSCOREs of 42.5% and 58.3%, respectively. PerplexityAI, which uses a commercial search engine and thus should have a perfect FACTSCORE if directly copying the text from the correct Wikipedia page, attains a FACTSCORE of 71.5%. We provide a qualitative analysis of its error cases in the last paragraph of this section.

ChatGPT and PerplexityAI often abstain from answering which presumably improves their factual precision. InstructGPT rarely abstains from answering, likely because it is not trained to do so.

Irrelevant facts either (a) have dependencies on previous facts in a generation that turn out to be unsupported, or (b) are irrelevant to the prompt independent from other facts in a generation (examples in Appendix A.4). We find that (b) rarely happens with InstructGPT and ChatGPT but happens considerably with PerplexityAI, because PerplexityAI often directly copies search results even if they are largely irrelevant to the input prompt. This is in agreement with a concurrent work from Liu et al. (2023a) that shows generative search engines like PerplexityAI copy incorrect search results and generate text that is irrelevant to the input query.

| Category | % | Example |
|---|---|---|
| Single-sentence contradiction (words) | 33.3 | `Gen` On November 25th, 2023, Glover Teixeira became an American citizen. `Wiki` In November 2020, Teixeira became an American citizen.
`Gen` [Eric Hacker] was named the International League Pitcher of the Year. `Wiki` [Eric Hacker] was named the IL Pitcher of the Week. |
| Single-sentence contradiction (beyond words) | 10.0 | `Gen` William Waldegrave's grandfather was James II and VII. `Wiki` His father's title was created ... for the diplomat and ambassador James Waldegrave, 1st Earl Waldegrave, whose grandfather was James II and VII.
`Gen` She has appeared in several successful films such as (...) and Zero (2018). `Wiki`: Zero was a commercial failure. |
| Page-level contradiction | 23.3 | `Gen` Some of [Julia Faye's] notable films include ... "Cleopatra" (1934). `Comment` No mention of *Cleopatra* on the *Julia Faye* page, and no mention of *Julia Faye* on the *Cleopatra* page.
`Gen` [Kang Ji-hwan] has donated money to various charities and organizations over the years. `Comment` No such mention on the *Kang Ji-hwan* page. |
| Subjective | 16.7 | `Gen` His achievements, as an actor and as a cultural force, will surely prove to be as heroic as those of the characters he portrayed. `Wiki` Culture writer Steve Rose, in The Guardian, wrote: "Chadwick Boseman began his career playing African American icons and pioneers; he ends it as one himself. His [...] achievements, as an actor and as a cultural force, will surely prove to be as heroic as those of the characters he portrayed." |
| Fact is irrelevant | 3.3 | `Gen` [Zamfir Arbore]'s life is not well-documented, and there is little information available about him. |
| Wiki is inconsistent & wrong | 3.3 | `Gen` Kick (2014) that brought [Sajid Nadiadwala] various debutant director awards. `Wiki` 2015, IIFA Award for Debut Director, Kick. (...) Kick brought him various debutant director awards. `Comment` The first text is from a table that indicates he won one award (accurate). The second is inaccurate, incorrectly citing a news article. |
| Annotation error | 10.0 | `Gen` [Zamfir Arbore] was part of the staff of Românul. `Wiki` The Românul staff came to include Zamfir Arbore. `Comment` Mentioned in the *Românul* page but not in the *Zamfir Arbore* page. |

Table 2: Categorization of precision errors (`Not-supported`) from PerplexityAI (Section A.5). `Gen` indicates the generation from PerplexityAI, and `Wiki` indicates evidence text from Wikipedia. `Comment` indicates our comments.

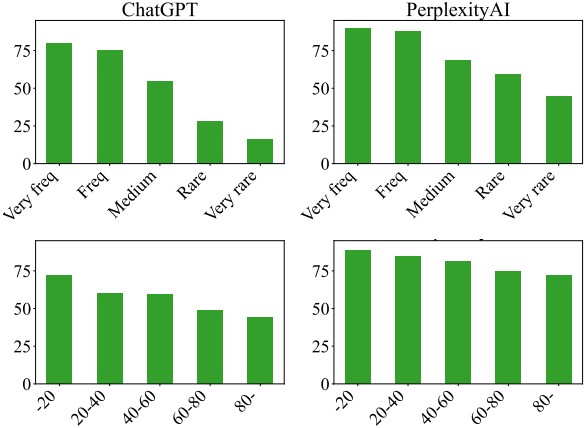

Figure 2: FACTSCORE across varying frequency levels of human entities (**top**) and relative positions in a generation (**bottom**). FACTSCOREs are lower as the rarity of the entities increases and the position of the fact is later.

**Error rates are higher for rarer entities.** Figure 2 (top) shows factual precision over varying frequency levels of topic entities (humans) in the pretraining corpora (see Appendix A.1). There is a notable decrease in FACTSCORE as the rarity of entities increases, consistently across all LM$_{SUBJ}$s. This is in agreement with Kandpal et al. (2022) and Mallen et al. (2023) which show that short question answering (QA) accuracy is highly correlated with the entity frequencies in the pretraining data. However, in contrast to Kandpal et al. (2022)

and Mallen et al. (2023) who report QA accuracy of models with retrieval is robust to the rarity of entities, FACTSCORE of PerplexityAI still significantly drops as entities are rarer: a relative drop of 50% and 64% observed at the atomic-level and sentence-level, respectively.

**Error rates are higher for facts mentioned later in the generation.** Figure 2 (bottom) reports factual precision over relative positions in a generation. Across all LMs, the later part of the generation has significantly worse precision. This is likely because (a) information mentioned earlier is more frequently mentioned in the pretraining data (e.g., nationality, profession), and (b) error propagation affects the later part of the generation. This also implies that evaluating LMs solely based on short answers may not provide an adequate assessment of their factual precision, as it fails to account for errors that arise in the later stages of generation.

**Qualitative analysis of `Not-supported`.** One of the surprising findings in our emprical analysis is that a FACTSCORE of PerplexityAI (71.5%) is lower than expected despite having access to the search engine. To better understand its errors, we categorize 30 random samples whose label is `Not-supported` (Table 2).

- Single-sentence contradiction: A single sentence from Wikipedia provides direct contradic-

tion to the generation, either at a word level (numbers, dates, or entities) or beyond.

- Page-level contradiction: Errors found after reading the entire page, often because a fact that should have been mentioned in Wikipedia if true is missing, e.g., whether the subject appears in a particular film.

- Subjective: Generation is subjective, often because PerplexityAI copies subjective text from Wikipedia, e.g., directly copying a quote from a journalist without realizing it.

- Fact is irrelevant: Generation is irrelevant to the subject due to a search error.

- Wiki is inconsistent & wrong: In the example, Wikipedia indicates that the subject won one award from the film Kick, but also includes text that they won multiple awards from Kick, which is inaccurate and cited a news article that does not support the claim.

- Annotation error: Annotators assign incorrect labels, typically because the information is not mentioned in the subject's Wikipedia page (likely because it is insignificant).

We also find that, although PerplexityAI provides citations to the references, citations have little correlation with factual precision. 36.0% and 37.6% of supported and unsupported sentences have citations, respectively. Together with independent findings from Liu et al. (2023a), this indicates that commercial LMs that incorporate search and provide citations may not be as reliable as expected.

More analysis is provided in Appendix A.5.

## 4 Estimating FACTSCORE for Automatic Evaluation

Human evaluation of factual precision is costly ($4 per generation) (Bohnet et al., 2022; Krishna et al., 2023) because validating every atomic fact against a large knowledge source is time-consuming, and one generation contains many (26–41) atomic facts. This prevents LM developers and practitioners from evaluating the factual precision in long-form generation of a new $\text{LM}_{\text{SUBJ}}$ at scale. In this context, we introduce a model that **estimates** FACTSCORE. This estimator takes a set of generations and automatically computes a FACTSCORE, and can be applied to any $\text{LM}_{\text{SUBJ}}$.

We describe our model (Section 4.1) and demonstrate its accuracy against human evaluation (Sec-

tion 4.2). FACTSCORE estimated by our model is then used to evaluate twelve LMs (Section 4.3).

### 4.1 Model

Our estimator of FACTSCORE first breaks a generation into a series of atomic facts and then validates each against the given knowledge source. We find taking atomic facts generated by InstructGPT (used in data collection in Section 3.3) effective and close to human, consistent with findings from prior work (Chen et al., 2022). This section thus focuses on how to validate each atomic fact against a given knowledge source.

The validation is based on zero-shot prompting of an LM referred to as an $\textbf{LM}_{\textbf{EVAL}}$ to distinguish from an $\text{LM}_{\text{SUBJ}}$. Specifically, a *prompt*—whose construction methods differ across four variants— is fed into an $\text{LM}_{\text{EVAL}}$. The prediction is then made by comparing the conditional probability of `True` and `False` from the $\text{LM}_{\text{EVAL}}$. If the logit values are unavailable (e.g., commercial LMs like Chat-GPT), the prediction is made based on whether the generated text contains `True` or `False`.[6]
The four variants we consider are as follows.

**No-context LM** uses `<atomic-fact> True or False?` as a prompt, closely resembling Kadavath et al. (2022).[7]

**Retrieve→LM** retrieves passages from the given knowledge source and then prompts the $\text{LM}_{\text{EVAL}}$. It first retrieves $k$ passages, constructs the prompt by concatenating retrieved passages, the given atomic fact, and "`True or False?`", and feeds it to the $\text{LM}_{\text{EVAL}}$ to get the prediction.

**Nonparametric Probability (NP)** makes a judgment based on a nonparametric likelihood. It masks out each token in the atomic fact, computes its likelihood using a nonparametric masked LM (Min et al., 2023), averages probabilities over all tokens, and makes a prediction based on thresholding.

**Retrieve→LM + NP** is an ensemble of Retrieve→LM and NP which assigns `Supported` only if both methods assign `Supported`.

---

[6]In Appendix B.3, we compare with an alternative prompting that generates a question and compares the answer to it and the expected answer (Kryscinski et al., 2020; Wang et al., 2020; Gao et al., 2022; Manakul et al., 2023). We empirically find that our prompting performs better due to the lack of control over the questions being generated.

[7]In Appendix B.3, we also compare with Self-check LM, a concurrent work from Manakul et al. (2023). We do not include it in the main paper because it has strong restrictions, e.g., requires access to the $\text{LM}_{\text{SUBJ}}$ at evaluation time and cannot be applied to PerplexityAI with nondeterministic outputs.

| Evaluator | | retrv | SUBJ: InstGPT | | SUBJ: ChatGPT | | SUBJ: PPLAI | | ranking |
|---|---|---|---|---|---|---|---|---|---|
| | | | ER | FS | ER | FS | ER | FS | |
| | Human | | | 42.5 | | 58.3 | | 71.5 | |
| Trivial | Always Supported | | 57.5 | 100.0 + | 41.7 | 100.0 + | 28.5 | 100.0 + | ✗ |
| | Always Not-supported | | 42.5 | 0.0 − | 58.3 | 0.0 − | 71.5 | 0.0 − | ✗ |
| | Always Random | | 7.5 | 50.0 + | 8.3 | 50.0 − | 21.5 | 50.0 − | ✗ |
| I-LLAMA | No-context LM | ✗ | 7.1 | 49.6 + | 7.8 | 50.5 − | 34.7 | 36.8 − | ✗ |
| | NP | ✓ | 14.8 | 57.3 + | 13.7 | 72.0 + | **1.4** | 72.9 | ✓ |
| | Retrieve→LM | ✓ | 14.1 | 56.6 + | 17.1 | 75.4 + | **0.1** | 71.6 | ✗ |
| | Retrieve→LM + NP | ✓ | **1.4** | 41.1 | **0.4** | 58.7 | 9.9 | 61.6 − | ✓ |
| ChatGPT | No-context LM | ✗ | 39.6 | 82.1 + | 31.7 | 90.1 + | 3.3 | 74.8 | ✗ |
| | Retrieve→LM | ✓ | 5.1 | 47.6 + | 6.8 | 65.1 + | 0.8 | 72.3 | ✓ |
| | Retrieve→LM + NP | ✓ | 5.2 | 37.3 − | 4.7 | 53.6 | 8.7 | 62.8 − | ✓ |

Table 3: Results on **Error Rate (ER)** along with FACTSCOREs estimated by each model (**FS**). '*retrv*' indicates whether or not retrieval is being used, and '*ranking*' ✓ indicates whether the ranking between three $LM_{SUBJ}$s rated by the model is consistent to the ground truth ranking. + and − respectively indicate the estimation is an overestimation and an underestimation by more than 5% in absolute. **Red Bold** indicates the best (lowest) ER. See Appendix B.2 for the results in other metrics that consider individual judgments instead of aggregated ones.

We use LLAMA 7B trained on Super Natural Instructions (Inst-LLAMA, Touvron et al., 2023; Wang et al., 2022) and ChatGPT as an $LM_{EVAL}$, and Generalizable T5-based Retrievers (GTR, Ni et al. (2022)) for passage retrieval. See Appendix B.1 for more implementation details.

### 4.2 Evaluation of Estimators

**Metrics.** We report **Error Rate (ER)**—the difference between the ground truth and the estimated FACTSCORE—as well as whether the estimated FACTSCOREs preserve the ranking between three $LM_{SUBJ}$s. Appendix B.2 discusses results with other metrics that consider individual judgments instead of aggregated judgments. We use the data in Section 3.3 as evaluation data.

Results are reported in Table 3.

**Retrieval significantly helps.** Models that use retrieval are consistently better than No-context LM which either has a significantly high ER or does not preserve ranking between three $LM_{SUBJ}$s. This is likely because the $LM_{EVAL}$ has not memorized every factual information about the topic entity, thus benefiting from retrieval providing factual context. Nonetheless, just using Retrieve→LM may overestimate FACTSCORE, e.g., by up to 17% with Inst-LLAMA, when a $LM_{SUBJ}$ is InstructGPT or ChatGPT. In this case, ensembling Retrieve→LM and NP reduces an error rate by a significant margin. When a $LM_{SUBJ}$ is PerplexityAI, single methods (either Retrieve→LM or NP) give a low ER, and ensemble methods have a higher ER due to an underestimation of FACTSCORE.

**ChatGPT is not always the best.** Our results show that ChatGPT is not necessarily better than Inst-LLAMA. We investigate this further in Appendix B.3. In summary, ChatGPT is better at validating each individual atomic fact. However, most errors from ChatGPT are incorrectly assigning Supported to unsupported facts, overestimating FACTSCORE. In contrast, LLAMA+NP is not biased toward overestimation or underestimation of the factual precision, resulting in an aggregated factual precision to be closer to the ground truth. This is similar to the trade-off between system-level and segment-level correlations in summarization evaluation, which often produce different rankings (Bhandari et al., 2020; Deutsch et al., 2021).

**The best estimator depends on the $LM_{SUBJ}$.** While using retrieval is consistently better than No-context LM, the best variant of estimator depends on a $LM_{SUBJ}$: LLAMA+NP for InstructGPT and ChatGPT, and ChatGPT for PerplexityAI. Nevertheless, both evaluators give consistently correct ranking between three $LM_{SUBJ}$'s, and Section 4.3 show scores from two estimators are largely correlated across 10+ $LM_{SUBJ}$s (0.99 Pearson's $r$). We recommend users try both variants of our estimator when evaluating a new $LM_{SUBJ}$ and report their correlation.

### 4.3 Evaluation of New LMs

Our estimator allows evaluating factual precision of a large set of new LMs at scale with no human

| LM$_{\text{SUBJ}}$ | Base LM | Use other LMs | Open | Release |
|---|---|---|---|---|
| InstructGPT | ? | ? | ✗ | Nov 2022 |
| ChatGPT | ? | ? | ✗ | Nov 2022 |
| GPT-4 | ? | ? | ✗ | Mar 2023 |
| Alpaca {7B,13B,65B} | LLAMA | InstructGPT | ✓ | Mar 2023 |
| Vicuna {7B,13B} | LLAMA | ChatGPT | ✓ | Mar 2023 |
| Dolly 12B | Pythia 12B | N/A | ✓ | Mar 2023 |
| Oasst-pythia 12B | Pythia 12B | N/A | ✓ | Mar 2023 |
| StableLM-tuned 7B | StableLM-base | ChatGPT, GPT-4 | ✓ | Apr 2023 |
| MPT Chat 7B | MPT 7B | ChatGPT | ✓ | May 2023 |

Table 4: A set of twelve LMs evaluated in Section 4.3. All models are tuned for instruction following or chat. *Use other LMs* indicates whether the model is trained on any data that includes outputs of another model. *Open* indicates model weights are publicly available.

efforts. As a case study, we evaluate ten new LMs that came out within two months at the time of conducting experiments (Table 4). These LMs were evaluated on many benchmarks but not in factual precision of long-form generation since such evaluation is costly. We aim to provide new insights on these LMs by estimating FACTSCORE of their long-form generations.

### 4.3.1 Setup

We evaluate 10 recently-released LMs as shown in Table 4. **GPT-4** (OpenAI, 2023) is a multi-modal LM released by OpenAI available through an API. **Alpaca** (Taori et al., 2023) is based on LLAMA (Touvron et al., 2023) fine-tuned on the instructions data based on InstructGPT following the recipe from Wang et al. (2022). **Vicuna** (Chiang et al., 2023) is based on LLAMA fine-tuned on the outputs from ChatGPT available through ShareGPT.[8] **Dolly**[9] is Pythia 12B (Biderman et al., 2023) fine-tuned on DataBricks Dolly, human-written data created by Databricks.[10] **Oasst-pythia**[11] is Pythia 12B fine-tined on human-written data collected through Open Assistant.[12] **StableLM-tuned-alpha**[13] is based on StableLM-base-alpha[14] fine-tuned on the data used in the Alpaca data, DataBricks Dolly, the ShareGPT data, the GPT4All data (Anand et al., 2023) and Anthropic HH (Bai et al., 2022). **MPT Chat** is based on MPT 7B[15] fine-tuned on the ShareGPT data, the Alpaca data, Anthropic HH, HC3 (Guo et al., 2023), and Evol-Instruct.[16]

We prompt each LM$_{\text{SUBJ}}$ to generate biographies of 500 human entities as done in Section 3.3 but

---

[8]sharegpt.com  [9]dolly-v2-12b  [10]databricks.com
[11]oasst-sft-1-pythia-12b  [12]open-assistant.io
[13]StableLM-tuned-alpha-7b  [14]stablelm-base-alpha-7b
[15]mosaicml.com/blog/mpt-7b  [16]evol_instruct_70k

---

| LM$_{\text{SUBJ}}$ | % responding | #facts / res |
|---|---|---|
| GPT-4 | 88.2 | 60.8 |
| Vicuna 13B | 76.6 | 50.9 |
| Vicuna 7B | 91.0 | 45.6 |
| Oasst-pythia 12B | 100.0 | 39.7 |
| StableLM-tuned-alpha 7B | 66.6 | 38.0 |
| MPT Chat 7B | 88.8 | 37.3 |
| ChatGPT | 84.2 | 37.0 |
| InstructGPT | 99.8 | 27.7 |
| Dolly 12B | 100.0 | 24.6 |
| Alpaca 7B | 100.0 | 17.4 |
| Alpaca 65B | 100.0 | 17.1 |
| Alpaca 13B | 100.0 | 16.6 |
| Human | 88.8 | 29.0 |

Table 5: Statistics of 500 model-generated bios in our unlabeled data from 12 LMs as well as human-written bios. *% responding* indicates % of generations that do not abstain from responding. *#facts / res* indicates # of atomic facts per response. LMs are sorted based on # of facts per response. See Figure 3 for their FACTSCOREs.

with no overlap in entities. We additionally include InstructGPT, ChatGPT, and human-written biographies obtained through DBPedia. Human-written biographies were unavailable for 11% of entities which we consider as abstaining from responding. See Table 5 for their statistics. In total, we evaluate 6,500 generations from 13 subjects, which would have cost $26K if they were evaluated by humans.

### 4.3.2 Results

Figure 3 shows the ranking between 13 subjects provided by the two best variants of our estimator whose scores are largely correlated, e.g., having a Pearson's $r$ of 0.99. This evaluation allows a better understanding of these models, including:

- All LMs are substantially less factual than humans. This is in contrast to prior work that claims LMs approach human performance, even for complex tasks (Ding et al., 2022; Nori et al., 2023; Lee et al., 2023) even though the task of writing biographies is fairly easy.

- GPT-4 and ChatGPT are comparable in factual precision. However, as reported in Table 5, GPT-4 abstains from responding less (12% vs. 16%) and generates significantly more facts (61 vs. 37 per response).

- GPT-4 and ChatGPT are significantly more factual than public models.

- Within the same family of models that differ in sizes, there is a clear correlation between the model size and factual precision, e.g., Alpaca 65B > 13B > 7B, and Vicuna 13B > 7B.

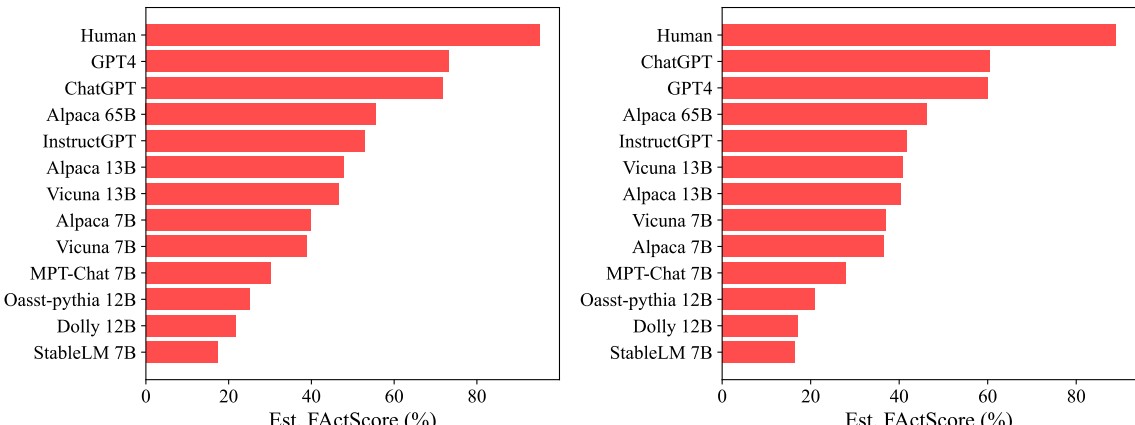

Figure 3: Ranking between 13 subjects (human and 12 LMs), rated by the two best variants of our estimator: ChatGPT (**left**) and LLAMA+NP (**right**), both with retrieval. Scores from two metrics have a Pearson's $r$ of 0.99. See Table 5 for % of responding and # of atomic facts per response of each LM. The variance in estimation based on different subsets of prompts is reported in Figure 5 of Appendix B.4.

- Alpaca and Vicuna achieve performance that is very close to each other within the same size of models, possibly because they share the same base model and similar training data. Nonetheless, as shown in Table 5, Vicuna generates significantly more atomic facts than Alpaca does (51 vs. 17 per response). Also, Alpaca never abstains from answering while Vicuna does.

- Within public models, there are large gaps in factual precision even when the model size is similar, e.g., within the 7B models, Alpaca and Vicuna ($\sim 40\%$) are more factual than MPT-Chat ($30\%$) and StableLM ($17\%$). Possible factors include the choice of the base LM, the data, and the training recipe (Hoffmann et al., 2022).

We highlight that this evaluation only considers factual precision, specifically in people biographies. A holistic evaluation of LMs should include other aspects of generations such as fluency, coherence, relevance, consistency and creativity, which is out of scope of this paper.

## 5 Conclusion and Future Work

We introduced FACTSCORE, a new evaluation of the factual precision of long-form generation from LMs that breaks a generation down into a series of atomic facts and computes a fraction of facts supported by a given knowledge source. We first performed extensive human evaluation, finding that commercial, state-the-art-art LMs—InstructGPT, ChatGPT, and search engine augmented, PerplexityAI—make a substantial amount of errors, e.g., having a FACTSCORE of 58% in

the case of ChatGPT. Since human evaluation is time-consuming and costly, we proposed a model that estimates FACTSCORE, allowing an automatic evaluation of FACTSCORE. We found our estimator based on retrieval over a knowledge source and competitive language models estimates FACTSCORE close to the ground truth, and showcased its application by evaluating 12 recently-released LMs that could have cost $65K if evaluated by humans and providing insights about them.

Within four months since its initial release, FACTSCORE has actively been used in subsequent work, evaluating factual precision of recently-proposed models (Ye et al., 2023; Sun et al., 2023; Malaviya et al., 2023; Dhuliawala et al., 2023). As future work, we suggest: (1) considering other aspects of factuality such as recall (coverage of factual information); (2) further improving the estimator for a better approximation of factual precision; and (3) leveraging FACTSCORE to correct model generations (briefly explored in Appendix C).

## Limitations

**Scope of FACTSCORE.** All of our experiments focus on people biographies and Wikipedia, because many LMs can generate biographies with objective and specific facts (rather than subjective and vague ones) and Wikipedia has a high coverage for them. FACTSCORE can be applied to a broader domain, e.g., text about recent events whose knowledge source can be a collection of news articles, or text about scientific findings whose knowledge source can be a collection of scientific literature. We present a proof of concept in Appendix B.5 and

leave further study for future work.

Due to the assumptions made in Section 3.1, FACTSCORE is not applicable when the facts are more nuanced, open-ended, and debatable (Chen et al., 2019; Xu et al., 2023) or with a knowledge source whose text frequently conflicts with each other (Wadden et al., 2022). Moreover, FACTSCORE may not be suitable for the human-written text that is nuanced and includes intentional or implicit deception.

**Limitation in our estimator.** While our estimator closely approximates humans and provides consistent ranking over a large set of LMs, it is not perfect in individual judgments, and the best variant depends on the degree of how close a generation is to human-written text and its linguistic complexity. Future work can investigate how the distribution of model generation affects the performance of the estimator and further improve the estimator.

**Beyond factual precision.** FACTSCORE focuses on factual precision—whether each piece of information in a generation is factually supported by a reliable source of knowledge—which is only one aspect of the broader factuality problem. For instance, FACTSCORE does not consider factual recall: the coverage of information in a generation. FACTSCORE does not penalize a model that abstains from responding too frequently or generates fewer facts, which can be unfair since there is an inherent trade-off between precision and recall. Moreover, the boundary between precision and recall is often blurry, e.g., it is possible that, even if every piece of information in a generation is supported, it misses a significant piece of information that should have been mentioned in order to be considered as correctly responding to the input prompt (example in Table 6). We leave a more holistic evaluation of factuality for future work, and recommend reporting FACTSCORE together with the % of abstention and the average number of atomic facts (as we did in Section 4.3).

## Acknowledgement

We thank Yizhong Wang for sharing Instruction-tuned LLAMA and Alpaca models with varying sizes, and for sharing feedback on the FACTSCORE Python package. We thank experts in Upwork for annotating the data, and Dhruba Ghosh, Jiacheng Liu and Zeqiu Wu for participating in pilot annotation and sharing feedback. We thank Akari

---

**Prompt:** Tell me a bio of Mary I of England.
**Generation:** Mary I was Queen of England and Ireland from July 1553 until her death. Mary was the daughter of King Henry VIII and his first wife, Catherine of Aragon. After her father annulled his marriage to Catherine, Mary's status was in doubt, and she was excluded from the line of succession. Her half-brother, Edward VI, succeeded their father in 1547, and Mary had to live with the humiliation of being declared illegitimate.

---

Table 6: An example whose factual precision is high but recall is low. The generation does not mention how Mary I of England got back to the line of succession and eventually became a queen.

Asai, Yanai Elazar, UW NLP members, UMass NLP members, FAIR lab members for feedback and discussion on the paper.

This research was supported by NSF IIS-2046248, NSF IIS-2202506, NSF IIS-2044660, ONR N00014-18-1-2826, ONR MURI N00014-18-1-2670, DARPA under Contract No. FA8650-23-C-7316, an Allen Distinguished Award, and gifts from AI2. The views, opinions and/or findings expressed are those of the author and should not be interpreted as representing the official views or policies of the Department of Defense or the U.S. Government. Sewon Min is supported by a J.P. Morgan fellowship, and Kalpesh Krishna was supported by the Google PhD Fellowship.

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

## A Details in Data Collection

### A.1 Sampling human entities

We sample 183 human entities to be annotated as follows. We first choose entities from Wikidata whose `instance of` is human and have corresponding Wikipedia pages. We then categorize entities based on two dimensions: frequency and nationality, resulting in 20 categories. We then sample entities uniformly at random over all categories.

**Frequency.** We compute `freqValue` as a maximum of the entity occurrence in Wikipedia provided by Kandpal et al. (2022) and the pageview count of the Wikipedia page following Mallen et al. (2023). We found using one of them could lead to an underestimate of frequency levels due to failure in entity linking or mismatch in the Wikipedia page title, and taking a maximum of them provides a reasonable solution. We then assign one of five categories: 'Very rare' if $freqValue \in [0, 10^2)$, 'Rare' if $freqValue \in [10^2, 10^3)$, 'Medium' if $freqValue \in [10^3, 10^4)$, 'Frequent' if $freqValue \in [10^4, 10^5)$, and 'Very frequent' if $freqValue \in [10^5, )$.

**Nationality.** We take `country of citizenship` from Wikidata and assign them one of four categories: 'North America', 'Europe & Middle East', 'Asia & Pacific' and 'Latin/South America & Africa'.

### A.2 Details in generating atomic facts

We break out a generation automatically by splitting a generation into sentences, and feeding each sentence to InstructGPT (`text-davinci-003`) with a series of instructions to further break it down to a series of atomic facts. The prompt to InstructGPT is provided in Table 15. Outputs from InstructGPT are used (1) to human experts for revision (Section 3.3) and (2) for model-based evaluators (Section 4). We find human experts split and merged atomic facts from InstructGPT for 18% and 34% of the cases, respectively.

### A.3 More details on annotator recruitment

We recruit freelancers through Upwork and pay 15–25 USD per hour. We recruit fact-checking experts—freelancers who mentioned fact-checking as their expertise—for Step 3. Every worker went through a qualification test of 2 hours and was tested to be highly qualified. We design one HIT to consist of three generations, one from each $LM_{SUBJ}$,

---

**Prompt:** Tell me a bio of Ylona Garcia.
**Sentence:** [Ylona Garcia] has since appeared in various TV shows such as ASAP (All-Star Sunday Afternoon Party), Wansapanataym Presents: Annika PINTAsera and Maalaala Mo Kaya.
- Ylona Garcia has appeared in various TV shows. `Supported`
- She has appeared in ASAP. `Supported`
- ASAP stands for All-Star Sunday Afternoon Party. `Supported`
- ASAP is a TV show. `Supported`
- She has appeared in Wansapanataym Presents: Annika PINTAsera. `Not-supported`
- Wansapanataym Presents: Annika PINTAsera is a TV show. `Irrelevant`
- She has appeared in Maalaala Mo Kaya. `Not-supported`
- Maalaala Mo Kaya is a TV show. `Irrelevant`

**Prompt:** Tell me a bio of John Estes.
**Sentence:** William Estes is an American actor known for his role on CBS police drama Blue Bloods as Jameson J̈amieR̈eagan.
- William Estes is an American. `Irrelevant`
- William Estes is an actor. `Irrelevant`
- William Estes is known for his role on CBS police drama Blue Bloods. `Irrelevant`
- William Estes' role on Blue Bloods is Jameson "Jamie" Reagan. `Irrelevant`

Table 7: Examples that contain `Supported`, `Not-supported` and `Irrelevant`. Sentences in bullet points indicate atomic facts.

---

for one prompt, because we find it saves annotation time in total. 10% of the HITs have two workers assigned to calculate the agreement rate; the rest have one worker assigned. The agreement rates are 96%, 90% and 88% for InstructGPT, ChatGPT and PerplexityAI, respectively. Appendix A.5 discusses disagreement cases in more detail. The full instructions and the interface are provided in Figure 6 and Figure 7, respectively.

### A.4 Examples in annotated data

Table 7 provides examples of the human-annotated data, each atomic fact with an assigned label. `Supported` and `Not-supported` respectively indicate Wikipedia supports the fact and does not support the fact (either contradicts or does not contain any evidence). `Irrelevant` indicates the fact is irrelevant to the input prompt, which can further be divided into two cases: (1) the fact depends on other facts because it expands previous facts in a generation, and such other facts are `Not-supported`, e.g., in the first example in Table 7, and (2) the entire sentence is irrelevant to the prompt, independent from other facts in a generation, e.g., the second example in Table 7. The second case rarely happens with InstructGPT and ChatGPT, but happens considerably with PerplexityAI, i.e., 24.7% of generations of PerplexityAI have ≥ sentences marked as irrelevant without dependencies to other facts, compared to 0.5% and

| Category | % | Example |
|---|---|---|
| Different interpretations of the factual information | 21 | `Gen` Gerhard Fischer is an inventor. `Wiki` Gerhard Fischer (inventor). ... was first patented by Dr. Gerhard Fischer in 1931. A metal detector had been invented some forty years earlier (1881) by Alexander Graham Bell ... 
 `Gen` Chadwick Boseman was a producer. `Comment` Chadwick Boseman is not known as a producer, but produced one music video. |
| Inferred (not directly mentioned but highly likely) | 16 | `Gen` Leach has since become a member of the England Test team. `Comment` Leach is a member of the England Test team, but since when is less clear. |
| Depends on how strict in judging the correctness | 11 | `Gen` He made his Test debut for England in March 2018. `Wiki` On 16 March 2018, he was called up to England's Test squad (...) He made his debut in the second Test in Christchurch. 
 `Gen` The building was the first LEED-certificated building in Edmonton. `Wiki` (..) became the first project in the City of Edmonton to achieve a LEED Gold status. |
| Subjective | 21 | `Gen` Chadwick Boseman became an African American pioneer. `Wiki` Culture writer Steve Rose, in The Guardian, said that Boseman's career was revolutionary and he "leaves behind a gamechanging legacy" (...) Rose wrote: "Chadwick Boseman began his career playing African American icons and pioneers; he ends it as one himself." |
| Wikipedia not consistent | 5 | `Gen` [Tim Fischer] was an Ambassador to the Holy See from 2009 to 2012. `Wiki` ... was later Ambassador to the Holy See from 2009 to 2012. (...) Australian Ambassador to the Holy See 2008–2012 `Comment` The plain text and the table of the *Tim Fischer* page as well as the *Australian Ambassador to the Holy See* page are inconsistent in his start year. |
| Two different entities | 5 | `Comment` Carlos J. Alfonso vs. Carlos Alfonso |
| Mistakes in annotation | 21 | `Gen` Jack Leach is a left-handed batsman. `Comment` mentioned in the *England cricket team* page, Table *Current Squad*. |

Table 8: Categorization of disagreement cases. `Gen` indicates the generation from PerplexityAI, and `Wiki` indicates evidence text from Wikipedia. `Comment` indicates our comments.

1.3% in InstructGPT and ChatGPT, respectively. This is because PerplexityAI often directly copies search results even if they are largely irrelevant to the input prompt. This is in agreement with a concurrent work from Liu et al. (2023a) that shows generative search engines like PerplexityAI copy incorrect search results and generate text that is irrelevant to the input query.

### A.5 Qualitative Analysis

**Analysis of disagreement cases.** We analyze the cases where two annotators assigned to a same generation disagree on a precision label for the same atomic fact. Categorization is provided in Table 8. The 70% is due to an inherent debatability on whether or not the fact is supported by a given source of knowledge, not satisfying Assumption 2 in Section 3.1. This is because there can be multiple interpretations of a fact, it is debatable whether or not an information can be inferred from a piece of text, or the atomic fact is subjective. For instance:

- `Gerhard Fischer is an inventor`: Gerhard Fischer is widely known as an inventor of a metal detector, and even the title of the Wikipedia article is *"Gerhard Fischer (inventor)"*. However, it later turns out that he did not invent a metal detector; rather, he commercialized it.

- `Chadwick Boseman was a producer`: Chadwick Boseman is widely known as another profession (singer) and there is no text that mentions him as a producer. However, he produced one music video.

Nonetheless, since our agreement rate is fairly high (91%), we think such cases are rare in our particular domain of people biographies. We include more discussion on other domains that such cases may be more frequent in the Limitation section.

**Coverage of English Wikipedia.** While factual prediction is inherently a function of a knowledge source given as part of the input, a potential concern is how representative using English Wikipedia as a knowledge source for evaluating people biographies with respect to its coverage. For instance, it is possible that, especially for rare entities, the coverage of information in Wikipedia is not high enough, and LMs may be penalized by generating information that is true even if not supported by Wikipedia (i.e., supported by other sources on the web).

To quantify the effect, we randomly sample 30 unsupported facts from ChatGPT on people whose categories are either 'rare' or 'very rare', and then validate them against the entire web. We found 10% (3 out of 30 facts) are in fact supported, even though they are not supported in Wikipedia. An example is *[Hibo] Wardere published her memoir titled "Cut:*

*One Woman's Fight Against FGM in Britain Today"* which is not mentioned in Wikipedia but is found from Google Books.

Nonetheless, we found that Wikipedia has a high coverage and mentions most of the important information that we were able to find from any other sources on the web. This is in agreement with prior work that treated Wikipedia as a general knowledge source under the same reason (Chen et al., 2017; Petroni et al., 2021).

## B  Details in Estimators

### B.1  Implementation details

As an $LM_{EVAL}$, we use the best open LM and the best commercial LM at the time of conducting experiments: LLAMA 65B (Touvron et al., 2023) and LLAMA 7B trained on Super Natural Instructions (Inst-LLAMA, Wang et al., 2022) as the former, and ChatGPT (OpenAI, 2022) as the latter. For computing nonparametric probabilities, we use a single-mask variant of NPM with BM25 as in the original paper (Min et al., 2023), and use 0.3 as a thresholding hyperparameter.

For passage retrieval, we use Generalizable T5-based Retrievers (GTR, a large variant), an unsupervised dense passage retrieval system (Ni et al., 2022). We restrict retrieved passages to be from the topic entity's page, and use $k = 5$. We find our estimator is not sensitive to the choice of a retrieval system (ablations provided in Appendix B.3). As a retrieval corpus, we use the English Wikipedia from 04/01/2023 which is around the time the data annotation was completed, and split each page into passages with up to 256 tokens.

**Additional baselines.** We also compare with **Self-check LM**, a method from a concurrent work by Manakul et al. (2023). Self-check LM needs multiple samples generated from the $LM_{SUBJ}$. It validates the given atomic fact by prompting $LM_{EVAL}$ conditioning on each generated sample,[17] making judgment (Supported or not) from each, and aggregates the results through a majority vote. This method assumes (1) the $LM_{SUBJ}$ is available at the time of evaluation and (2) the outputs from the $LM_{SUBJ}$ are nondeterministic, which makes it not applicable to PerplexityAI.

---

[17]Manakul et al. (2023) uses BERTSCore and a supervised question answering system instead of LM prompting, however, we find LM prompting to be significantly better.

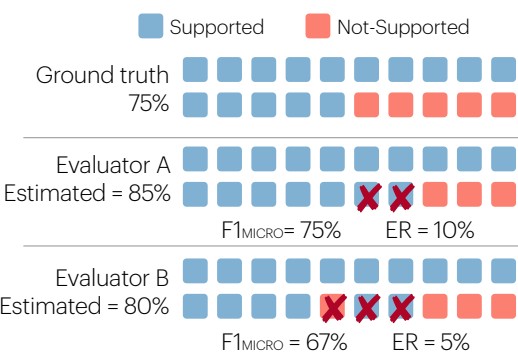

Figure 4: A case in which $F1_{MICRO}$ and Error Rate (ER) rank two evaluators differently. Evaluator A is better in $F1_{MICRO}$, and Evaluator B is better in ER.

### B.2  Segment-level vs. system-level evaluation

Besides how close the estimated FACTSCORE is to the ground truth FACTSCORE (**Error Rate**, as reported in Section 4), we also report $F1_{MICRO}$. $F1_{MICRO}$ evaluates how well the model validates each individual atomic fact, assuming oracle atomic facts (atomic facts by human experts) are given, and evaluates how good the estimator is in identifying facts that are not Supported (NS). Formally, let $\mathcal{G}$ and $\mathcal{P}$ be sets of atomic facts in a set of generations that have Not-supported as a ground truth label and as a predicted label, respectively. We define $F1_{MICRO}$ as follows.

$$\mathrm{P} = \frac{\mathcal{P} \cap \mathcal{G}}{\mathcal{P}}, \quad \mathrm{R} = \frac{\mathcal{P} \cap \mathcal{G}}{\mathcal{G}}, \quad \mathbf{F1}_{\mathrm{MICRO}} = \frac{2 \cdot \mathrm{P} \cdot \mathrm{R}}{\mathrm{P} + \mathrm{R}}$$

We call them MICRO because they consider individual decisions rather than aggregated estimation.

**ER vs. $F1_{MICRO}$.** $F1_{MICRO}$ cares about the individual decision, while ER cares about the aggregated estimation. An evaluator that has a high (better) $F1_{MICRO}$ but always overestimates or underestimates factual precision may have a higher (worse) ER, e.g., Evaluator A in Figure 4. Conversely, an evaluator that has a lower (worse) $F1_{MICRO}$ but is not biased toward overestimation nor underestimation may have a lower (better) ER, e.g., Evaluator B in Figure 4. Prior work in model-based evaluation mainly reports aggregated scores since the goal is a comparison between different systems being evaluated (Zhang et al., 2020; Rashkin et al., 2021; Gao et al., 2022) while we report both to see the relationship between two types of metrics. $F1_{MICRO}$ and ER are also closely related to *segment*-level and *system*-level correlations to human judgments respectively, which have been extensively used in

| Evaluator | retrv | LM$_{\text{SUBJ}}$ | | |
|---|---|---|---|---|
| | | InstGPT | ChatGPT | PPLAI |
| Always Supported | - | 0.0 | 0.0 | 0.0 |
| Always Not-supported | - | 71.4 | 58.3 | 30.9 |
| Random | - | 52.2 | 45.0 | 25.7 |
| No-context LM | ✗ | 61.2 | 52.2 | 31.4 |
| Self-check LM | ✗ | 66.0 | 48.4 | - |
| Retrieve→LM | ✓ | 78.7 | 61.9 | 51.1 |
| NP | ✓ | 70.0 | 56.6 | 51.4 |
| Retrieve→LM + NP | ✓ | **83.2** | **70.5** | **53.3** |

Table 9: Results in $\textbf{F1}_{\text{MICRO}}$ using Inst-LLAMA 7B as an LM$_{\text{EVAL}}$. '*retrv*' indicates whether or not retrieval is used. Self-check is not applicable to PerplexityAI whose outputs are semi-deterministic. **Bold** indicates the best performance.

| Evaluator | retrv | LM$_{\text{SUBJ}}$ | | |
|---|---|---|---|---|
| | | InstGPT | ChatGPT | PPLAI |
| ***LLAMA 65B*** | | | | |
| No-context LM | ✗ | 22.2 | 20.0 | 18.6 |
| Retrieve→LM | ✓ | 54.6 | 42.1 | 36.1 |
| Retrieve→LM + NP | ✓ | 80.1 | 67.1 | **55.1** |
| ***Inst-LLAMA 7B*** | | | | |
| No-context LM | ✗ | 61.2 | 52.2 | 31.4 |
| Retrieve→LM | ✓ | 78.7 | 61.9 | 51.1 |
| Retrieve→LM + NP | ✓ | **83.2** | **70.5** | 53.3 |
| ***ChatGPT*** | | | | |
| No-context LM | ✗ | 40.0 | 25.4 | 25.4 |
| Retrieve→LM | ✓ | **87.5** | **80.2** | **65.8** |
| Retrieve→LM + NP | ✓ | 86.6 | 77.8 | 60.8 |

Table 10: Ablation in $\textbf{F1}_{\text{MICRO}}$ on the choices of LM$_{\text{EVAL}}$. '*retrv*' indicates whether or not retrieval is used. **Bold** and **Red bold** indicate the best F1 within open-access LMs and commercial LMs, respectively.

developing evaluation metrics in machine translation (Ma et al., 2019; Thompson and Post, 2020) and summarization (Bhandari et al., 2020; Deutsch et al., 2021).

**Results.** Results on F1$_{\text{MICRO}}$ are reported in Table 9. Self-check LM outperforms no-context LM by 4–11%, which confirms findings from Manakul et al. (2023). However, both significantly underperform methods that use retrieval. This is in contrast to Manakul et al. (2023) that reports that Self-check without retrieval achieves performance that is close to that with retrieval, likely because the data in Manakul et al. (2023) contains more frequent entities. The fact that retrieval significantly helps is consistent with findings in Section 4.2 with an ER as a metric.

Adding NP improves Retrieve→LM by 2–9%, again consistent with findings in Section 4.2. This is likely because Retrieve→LM often makes incorrect predictions when there is a strong bias from an LM or there are distracting passages, and considering nonparametric probabilities makes the model more robust to these factors. For instance, given an unsupported fact `Samuel Oboh is Nigerian`, No-context LM, Self-check LM and Retrieve→LM predict `Supported` due to a strong name-nationality bias. NPM correctly predicts `Not-supported` based on a passage `Samuel Oboh ... is a Canadian architect, manager, ...`. It is also worth noting that this is different from findings in Section 4.2 that ChatGPT is not necessarily better than LLAMA+NP based on ER.

**Using a stronger LM$_{\text{EVAL}}$ significantly improves F1$_{\text{MICRO}}$.** Table 10 reports a comparison across different choices of an LM$_{\text{EVAL}}$. Within the same method, Inst-LLAMA 7B outperforms LLAMA 65B, and ChatGPT outperforms both. Using retrieval is critical across all models, e.g., the best no-context model based on ChatGPT is underperformed by all models with retrieval. Using NP helps LLAMA-based models but not ChatGPT, likely because ChatGPT is less affected by incorrect prior from the LM or distracting passages.

It is worth noting that these results are somewhat different from findings in Section 4.2 that ChatGPT is not necessarily better than LLAMA+NP. This is becauase, although ChatGPT is better in validating each individual atomic fact, most errors from ChatGPT are incorrectly assigning `Supported` to `Not-supported` facts, resulting in an overestimation of FACTSCORE. In contrast, LLAMA+NP is not biased toward overestimation or underestimation of the factual precision, resulting in an aggregated factual precision to be closer to the ground truth. This is similar to the trade-off between system-level and segment-level correlations in summarization evaluation (Bhandari et al., 2020; Deutsch et al., 2021).

### B.3 Ablations

**QA Prompting vs. TF Prompting** As described in Section 4.1, we use `True` or `False` as part of the prompt, so-called `TF` Prompting. An alternative is `QA` Prompting, which generates a question and the expected answer, obtains the answer for the generated question independent from the expected answer, and compares the expected answer

| Evaluator | LM$_{\text{SUBJ}}$ | | |
|---|---|---|---|
| | InstGPT | ChatGPT | PPLAI |
| Always Supported | 30.8 | 37.1 | 45.0 |
| Always Not-supported | 35.7 | 29.1 | 15.5 |
| Random | 50.5 | 50.2 | 43.2 |
| **QA** *Prompting* | | | |
| No-context LM | 56.5 | 48.8 | 32.5 |
| Self-check LM | 65.3 | 63.2 | - |
| Retrieve→LM | 65.3 | 58.2 | 47.3 |
| **TF** *Prompting* | | | |
| No-context LM | 57.3 | 55.3 | 41.7 |
| Self-check LM | 68.0 | 61.9 | - |
| Retrieve→LM | **78.9** | **71.4** | **69.2** |

Table 11: Results on F1$_{\text{MICRO}}$, comparing between the QA prompting and TF Prompting. We use Inst-LLAMA 7B as an LM$_{\text{EVAL}}$. Self-check is not applicable to PerplexityAI since PerplexityAI outputs are semi-deterministic. **Bold** indicates the best F1$_{\text{MICRO}}$.

| Retrieval | LM$_{\text{SUBJ}}$ | | |
|---|---|---|---|
| | InstGPT | ChatGPT | PPLAI |
| BM25 | 78.5 | 70.8 | 69.1 |
| GTR Large | 78.9 | **71.4** | **69.2** |
| GTR xLarge | **79.2** | 71.3 | 69.0 |

Table 12: Results on F1$_{\text{MICRO}}$, comparing different retrieval systems: BM25, GTR Large and GTR xLarge, all with Retrieve→LM based on Inst-LLAMA 7B. **Bold** indicates the best F1$_{\text{MICRO}}$.

and the predicted answer. This approach has been widely studied in the summarization literature and recent work in factual precision (Kryscinski et al., 2020; Wang et al., 2020; Gao et al., 2022; Manakul et al., 2023). Table 11 provides a comparison between two types of prompting. The TF approach significantly outperforms the QA approach, consistently over all methods. Our further analysis finds that this is due to generated questions often being overly vague or ambiguous. For instance, given a supported fact `Samuel Oboh is an architect`, the LM generates `What is Samuel Oboh's job?` as a question and `Architect` as an expected answer, and the obtained answer is `Vice President`. Although both `Architect` and `Vice President` are correct, they are not the same, thus the model incorrectly predicts `Not-supported`. Such cases make the model overpredict `Not-supported`, leading to many incorrect predictions.

**Impact of the choice of retrieval.** Table 12 compares Retrieve→LM methods based on a few passage retrieval systems, including BM25 (Lin et al., 2021), GTR Large and GTR xLarge. Results indi-

| Category | % |
|---|---|
| No direct evidence from retrieved passages | 70 |
| Distracted by other passages | 17 |
| Atomic fact is context-dependent | 7 |
| Wrong prediction even with the right passage | 3 |
| Annotation error | 3 |

Table 13: Categorization of 30 samples incorrectly predicted by Retrieve→LM based on ChatGPT.

cate that all retrieval systems are equally good and Retrieve→LM is not sensitive to the choice of the retrieval system.

**Qualitative analysis.** Table 13 categories errors made by Retrieve→LM based on ChatGPT, the evaluator with the best F1$_{\text{MICRO}}$. 70% of the errors are due to retrieved passages not providing direct evidence (either support or contradiction). These are difficult even for state-of-the-art retrieval systems and language models because validating facts often requires reading the entire page rather than a single passage, e.g., an actor not appearing in a particular film. 17% of errors are made because ChatGPT is being distracted by other passages, although it assigns a correct label if only a particular, correct passage is given.

### B.4 More details in evaluation of new LMs (Section 4.3)

**Variance in estimation.** Figure 5 reports FACTSCOREs estimated by two variants of our estimator as in Figure 3 but with 100 random subsets of the data. Specifically, we chose $N$ samples (out of 500) uniformly at random across 20 categories (defined in Appendix A.1) $M$ times and report the average and the standard deviation. We use $N = \{40, 100, 200\}$ and $M = 100$. Results indicate that the variance is overall low, preserving ranking between 13 subjects in most cases. As expected, the variance is lower as the sample size gets larger. Finally, the estimator based on ER based on LLAMA+NP (bottom) has an overall lower variance than the estimator based on ChatGPT (top).

### B.5 Feasibility in applying FACTSCORE to other domains

As mentioned in the Limitation section, our paper mainly evaluates on people biographies using Wikipedia. Evaluating the generalizability of FACTSCORE to other types of prompts and other domains is an avenue for future work.

As a proof of conept, we conduct small-scale

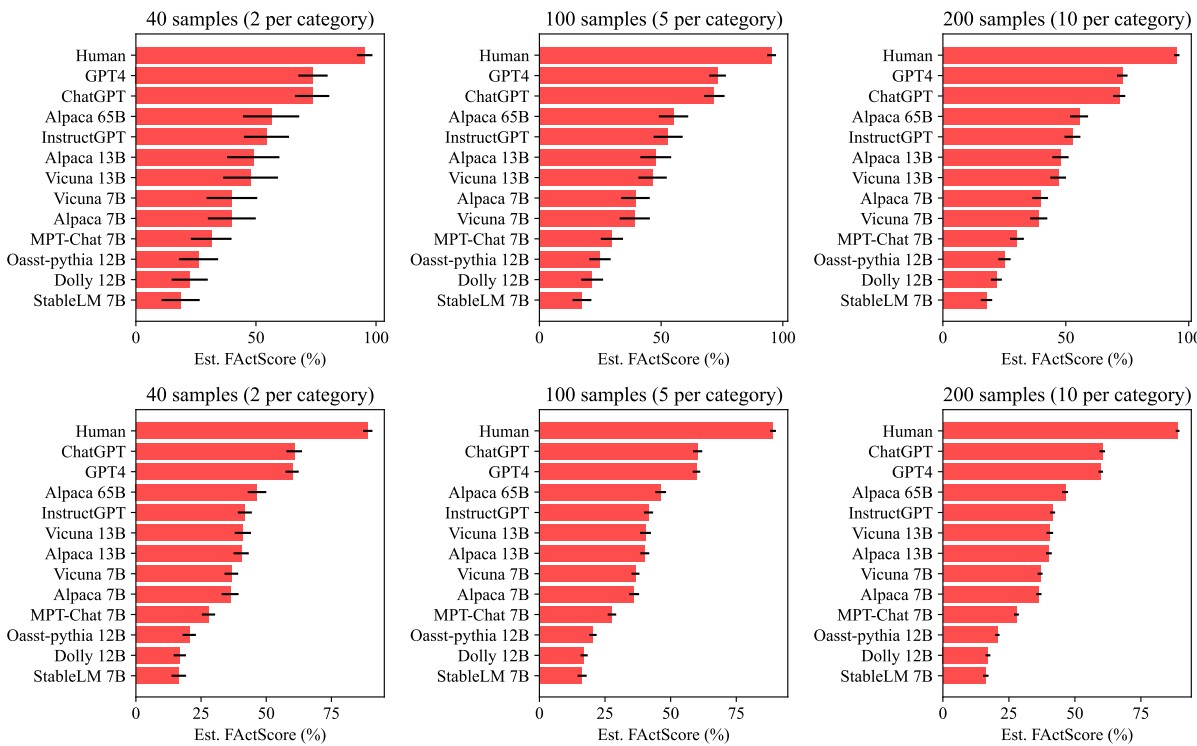

Figure 5: Impact of different subsets of random samples in prompts. The FACTSCOREs to 13 subjects (human and 12 LMs) are rated by the two best variants of our estimator: ChatGPT (**Top**) and LLAMA+NP (**Bottom**), both with retrieval. The variance is overall low, and is lower as the sample size gets larger and with LLAMA+NP (bottom) than with ChatGPT (top).

studies in the NLP domain. We first manually write 10 prompts asking about NLP papers: *Tell me a summary of* `<paper-title>`, and then obtain responses from ChatGPT. Next, we run FACTSCORE against an ACL anthology as a knowledge source. Finally, we compute an error rate (ER)—a difference between humans' validation (labeled by authors) and the model's validation—as we do in Section 4. The ER is 7.41 (FACTSCORE from humans being 66.20, and FACTSCORE from the model being 73.61), which is comparable to ER values in people bios shown in Table 3.

This suggests that FACTSCORE can generalize beyond people biographies. However, since this is a very small-scale experiment, we strongly encourage future research to explore the generalizability of FACTSCORE to more domains at scale.

## C Editing Experiments

Our experiments in Section 4 focuses on automatically identifying factual precision errors in long-form generations by language models. Can these labels be used to actually correct errors in the long-form generations? In this section, we perform a preliminary exploration of methods to edit long-form LM generations to reflect factually correct information. We assume we have access to the human-annotated set of FACTSCORE labels, and measure how good models are at editing incorrect sentences. In other words, we evaluate our editor models independent of the errors arising from the estimator.

### C.1 Methods

We adopt a similar set of methods as Section 4.1 for our editing models. All methods below use four exemplar examples for in-context learning which were sampled from our dataset and removed for subsequent analysis. For all methods, we use OpenAI's ChatGPT (OpenAI, 2022) as the base language model due to its generative capabilities.

**No-context LM**. We feed language models the prompt `Input: <sentence> Edit:` and ask it to edit the text, without any retrieved context.

**Retrv→LM**. To assist an editor model, we use a passage retrieval system to find supporting evidence from an external knowledge source (Wikipedia in our case). Our retrieval pipeline is

identical to Appendix B.1, but uses 3 retrieved passages instead of 5 due to context length restrictions.

**+ Atomic Facts**. Additionally, we explore whether adding atomic facts and their labels assist a model with fine-grained editing. Specifically, after the input sentence we add information to the prompt of the form `Fact 1 (True/False): <atomic fact 1> Fact 2 (True/False): <atomic fact 2>` ... This data is also provided in the exemplars.

**Non-edit baselines**. Finally, we add some trivial baselines to lower-bound our editing metrics. Specifically, we measure the performance of input copying (no edits), as well as an editor with random token dropping / replacement on a random 25% subset of tokens.

## C.2  Evaluation

In our data collection process (Section 3.3), along with our verification data we also collected gold-standard human written edits. Let $X = x_1, ...x_{N_X}$ be the input sentence and $G = g_1, ...g_{N_G}$ be the gold edited sentence. We evaluate the quality of the model-generated edit ($E = e_1, ..., e_{N_E}$) using three automatic metrics,

(1) **Error Localization** (ErrLoc): Our first metric measures how well the editor identifies errors within the input sentence. Specifically, we first create a "token preservation string", marking token $x_i$ in the input sentence $X$ as "Preserved" or "Not Preserved". We then compute the macro-averaged F1 score between the token preservation strings derived from the gold edit and the model-generated edit. We remove stopwords, punctuation and lowercase all words before performing this calculation. To equally weigh every sentence, F1 scores are independently computed for each sentence before a final averaging.

(2) **Edit Correctness** (EditCorr): Our second metric assesses the quality of the additional tokens added by the model-generated edit. Specifically, we check the token-level F1 score (Rajpurkar et al., 2016) comparing the new tokens added by the gold edit $G$ and the new tokens added by the model-generated edit $E$. More concretely,

$$N_{\text{common}} = \sum_{e_i \in E, e_i \notin X} e_i \in G$$

$$\text{precision} = N_{\text{common}} \ / \ ||\{e_i \in E, e_i \notin X\}||$$

$$\text{recall} = N_{\text{common}} \ / \ ||\{g_i \in G, g_i \notin X\}||$$

$$\text{EditCorr (F1)} = \text{HM}(\text{precision}, \text{recall})$$

where $|| \cdot ||$ is the set cardinality and HM denotes a harmonic mean. For this metric, we discard data points where the gold edit did not add new tokens. Similar to ErrLoc, we also remove stopwords, remove punctuation and lowercase strings before calculating EditCorr scores.

(3) **SIM alignment** (SimAl): Finally, due to the large output space of possible edits, we also adopt a metric which rewards paraphrases of the gold edits. We use semantic similarity embeddings from Wieting et al. (2022) which map paraphrases to a similar part of a vector space. We check the similarity between the model edit $E$ and the gold edit $G$, normalizing it by the similarity between $G$ and the original input $X$.[18] Specifically,

$$\text{Sim} = \max\left(0, \frac{s(G, E) - s(G, X)}{1 - s(G, X)}\right)$$

where $s(A, B)$ is the semantic similarity score (normalized to $[0, 1]$) from the model in Wieting et al. (2022). Intuitively, this metric measures how much closer $G$ and $E$ are compared to $G$ and $X$.

## C.3  Results

We present our editing results in Table 14. Overall, we find that:

**All editing models perform better than trivial lower bounds.** Overall, we find that all editor models outperform lower-bound baselines like random noise. This even happens in the no-context LM setting, where ChatGPT is editing its own output (or search engine augmented Perplexity AI's outputs), but can still perform non-trivial corrections (6.8 ErrCorr for ChatGPT correcting its own outputs vs 0.1 for a random noise editor baseline).

**Retrieval significantly helps with editing performance.** Across all base language models and metrics, augmenting the editor with retrieved paragraphs boosts performance (6.8 → 16.8 ErrCorr, 4.0 → 9.5 SimAl for ChatGPT correcting its own outputs). We hypothesize that the internal parametric knowledge in ChatGPT has insufficient information about the topic (as we also observed in Section 3.4) to perform fine-grained editing, and using external knowledge from Wikipedia greatly simplifies error localization and correction. This also corroborates with our findings in Section 4.2.

---

[18]We avoid taking the vector differences between the original / edited text since edit vectors (Guu et al., 2018) were not explicitly modeled in Wieting et al. (2022).

|  | InstructGPT | | | ChatGPT | | | PerplexityAI | | |
|---|---|---|---|---|---|---|---|---|---|
| Editor | ErrLoc | ErrCorr | SimAl | ErrLoc | ErrCorr | SimAl | ErrLoc | ErrCorr | SimAl |
| Input copying | 37.1 | 0.0 | 0.0 | 38.8 | 0.0 | 0.0 | 45.6 | 0.0 | 0.0 |
| 25% random noise | 44.1 | 0.1 | 0.5 | 45.5 | 0.1 | 0.4 | 45.2 | 0.0 | 0.3 |
| *ChatGPT* | | | | | | | | | |
| No-context | 49.0 | 8.5 | 6.2 | 45.3 | 6.8 | 4.0 | 48.3 | 6.2 | 4.1 |
| No-context + atomic facts | 58.7 | 12.7 | 10.5 | 53.4 | 10.0 | 6.6 | 56.0 | 9.6 | 6.1 |
| Retrv→LM | 52.6 | 21.8 | 15.7 | 43.9 | 16.8 | 9.5 | 46.3 | 13.5 | 6.8 |
| Retrv→LM + atomic facts | **65.4** | **30.4** | **25.5** | **63.5** | **28.3** | **19.3** | **62.4** | **23.6** | **15.9** |

Table 14: Results after automatic editing with ChatGPT assuming ground truth verification labels. All editors perform better than trivial lowerbound baselines, and using retrieval and atomic fact labels boosts editing performance. Details of automatic metrics (ErrLoc, ErrCorr, SimAl) are defined in Section C.2.

**Atomic fact labels improve error localization and improve editing performance.** Across all base language models (with or without retrieval) we observe that providing fine-grained atomic fact labels improves editing performance ($16.8 \rightarrow 28.3$ ErrCorr, $9.5 \rightarrow 19.3$ SimAl for ChatGPT correcting its own outputs). Fine-grained fact correctness labels help the editor easily identify problematic tokens, as seen by the consistent improvements in ErrLoc scores ($43.9 \rightarrow 63.5$ for ChatGPT correcting itself). We hypothesize atomic facts help guide the editor with its editing process (for instance, perform a more targeted search in the retrieved paragraphs), resulting in ErrCorr improvements. We also find that atomic fact labels reduces the frequency of editor copying the input verbatim or saying *The input has no errors* from 37.3% to 3.9%.

**PerplexityAI outputs are the hardest to edit**. Overall, we find the highest editing success for InstructGPT, followed by ChatGPT and the least success for Perplexity AI. We hypothesize this is because PerplexityAI already uses a search engine, so errors are much more subtle as extensively discussed in Appendix A.5.

Please breakdown the following sentence into independent facts: He made his acting debut in the film The Moon is the Sun's Dream (1992), and continued to appear in small and supporting roles throughout the 1990s.
- He made his acting debut in the film.
- He made his acting debut in The Moon is the Sun's Dream.
- The Moon is the Sun's Dream is a film.
- The Moon is the Sun's Dream was released in 1992.
- After his acting debut, he appeared in small and supporting roles.
- After his acting debut, he appeared in small and supporting roles throughout the 1990s.

Please breakdown the following sentence into independent facts: He is also a successful producer and engineer, having worked with a wide variety of artists, including Willie Nelson, Tim McGraw, and Taylor Swift.
- He is successful.
- He is a producer.
- He is a engineer.
- He has worked with a wide variety of artists.
- Willie Nelson is an artist.
- He has worked with Willie Nelson.
- Tim McGraw is an artist.
- He has worked with Tim McGraw.
- Taylor Swift is an artist.
- He has worked with Taylor Swift.

Please breakdown the following sentence into independent facts: In 1963, Collins became one of the third group of astronauts selected by NASA and he served as the back-up Command Module Pilot for the Gemini 7 mission.
- Collins became an astronaut.
- Collins became one of the third group of astronauts.
- Collins became one of the third group of astronauts selected.
- Collins became one of the third group of astronauts selected by NASA.
- Collins became one of the third group of astronauts selected by NASA in 1963.
- He served as the Command Module Pilot.
- He served as the back-up Command Module Pilot.
- He served as the Command Module Pilot for the Gemini 7 mission.

Please breakdown the following sentence into independent facts: In addition to his acting roles, Bateman has written and directed two short films and is currently in development on his feature debut.
- Bateman has acting roles.
- Bateman has written two short films.
- Bateman has directed two short films.
- Bateman has written and directed two short films.
- Bateman is currently in development on his feature debut.

Please breakdown the following sentence into independent facts: Michael Collins (born October 31, 1930) is a retired American astronaut and test pilot who was the Command Module Pilot for the Apollo 11 mission in 1969.
- Michael Collins was born on October 31, 1930.
- Michael Collins is retired.
- Michael Collins is an American.
- Michael Collins was an astronaut.
- Michael Collins was a test pilot.
- Michael Collins was the Command Module Pilot.
- Michael Collins was the Command Module Pilot for the Apollo 11 mission.
- Michael Collins was the Command Module Pilot for the Apollo 11 mission in 1969.

Please breakdown the following sentence into independent facts: He was an American composer, conductor, and musical director.
- He was an American.
- He was a composer.
- He was a conductor.
- He was a musical director.

Please breakdown the following sentence into independent facts: She currently stars in the romantic comedy series, Love and Destiny, which premiered in 2019.
- She currently stars in Love and Destiny.
- Love and Destiny is a romantic comedy series.
- Love and Destiny premiered in 2019.

Please breakdown the following sentence into independent facts: During his professional career, McCoy played for the Broncos, the San Diego Chargers, the Minnesota Vikings, and the Jacksonville Jaguars.
- McCoy played for the Broncos.
- McCoy played for the Broncos during his professional career.
- McCoy played for the San Diego Chargers.
- McCoy played for the San Diego Chargers during his professional career.
- McCoy played for the Minnesota Vikings.
- McCoy played for the Minnesota Vikings during his professional career.
- McCoy played for the Jacksonville Jaguars.
- McCoy played for the Jacksonville Jaguars during his professional career.

Please breakdown the following sentence into independent facts

Table 15: A prompt given to InstructGPT to generate atomic facts for a given sentence. Model generated atomic facts were revised by human editors.

You will be given up to three pieces of text as a response to a user query. Make each text factually correct based on Wikipedia.

Step 1: Skim 1

Read the given user query and the text to get familiarized with the topic.

Step 2: Verify & Edit 1

Verify and edit the text sentence-by-sentence. Each sentence is paired with a series of facts embedded in the sentence. You will go through three steps for each sentence.

1. For each fact, verify whether it is factually correct based on Wikipedia. Navigate different Wikipedia pages extensively, and assign one of three labels:
   - *Supported*: you found the Wikipedia text that indicates that the fact is definitely correct.
   - *Not-supported*: not "Supported", either because you found the Wikipedia text that indicates the fact is incorrect, or the fact is unverifiable.
   - *Irrelevant*: the fact is not related to the user query anymore, and will have to be deleted regardless of its correctness. ❶ **Make sure to contextualize each fact with the original sentence**, e.g., in the sentence Marie Curie won the Nobel Prize in Chemistry while she was in Sorbonne University, the fact Marie Curie was in Sorbonne University is only true when Marie Curie was in Sorbonne University when she won the Nobel Prize. If Marie Curie was in Sorbonna University but not while she won the Nobel Prize, you should choose *Not-supported*.
   ❶ **Make sure to focus on differences between each fact**. For instance, given two facts He won the Grammy Award and He won the Grammy Award in 2013, consider the second fact focuses on the fact that He won an award *in 2013*. Thus, if he won the Academy award not the Grammy award in 2013, mark the first fact as *Not-supported* and the second fact as *Supported*, and edit the sentence by replacing Grammy to Academy.
   ❶ **Correct partially-true, half-true or misleading facts as well, even if they are not literally false.** For instance, if the given fact is Liz Taylor and Richard Burton divorced. whereas the truth is they divorced and re-married, then you should select *Not-supported* and edit the sentence to clarify their re-marriage, unless the later sentence clarifies their remarriage.
   ❶ If the edits made in the earlier facts (or sentences) make th later facts (or sentences) irrelevant or duplicated, the later facts should be labeled as *Irrelevant*. Examples:
     - Brad Johnson's debut film is Always, which is Steven Spielberg's 1989 film. is paired with three facts, (1) Brad Johnson's debut film is Always. (2) Always is directed by Steven Spielberge. (3) Always is released in 1989. Here, (1) is false, because Brad Johnson's debut film is Nam Angels. Then, (2) and (3) should be marked irrelevant, regardless of they are true or false, because they are not related to the topic of the sentence (Brad Johnson's debut film) anymore. And you should edit the sentence to include the fact that Brad Johnson's debut film is Nam Angels.
     - If the later sentence after this sentence is Brad Johnson also appeared in Nam Angels, then this fact should be labeled Irrelevant, even if it is factually true, because this fact has already appeared in the edited version of the earlier sentence, and keeping this sentence will lead to duplication.
2. For a *Supported* fact, select a supporting sentence: a sentence that supports that the corresponding fact is true. You can do so when sentences of the Wikipedia article you are currently viewing are activated for you to click. You can only choose one sentence for each fact.
   ❶ If there is no single sentence that supports the fact, and rather the whole section or the whole article is necessary to make a decision, click the section title or the article title instead of the sentence. However, only do so when necessary.
   ❶ You can choose the list items or the table as a supporting sentence. However, please avoid this if there is *raw text* that supports the fact.
3. If any of the fact in the sentence is *Not-supported* or *Irrelevant*, edit the sentence to make it factually correct based on Wikipedia. You can replace entities or numbers to the correct ones, or delete facts in case replacing is not easy. **Remember, correct factual errors only. Do not make a correction for writing style, etc.**

Step 3, Step 4, Step 5 & Step 6

Repeat Step 1 & Step 2 with the second piece of text. Some of these steps may be skipped. Remember, the decision of different paragraphs should not depend on each other.

Tips

- Navigate Wikipedia extensively! The default page is only an estimate of the most relevant page, but you are supposed to navigate the entire Wikipedia and find information from other pages if needed. Use hyperlinks or a "search" function to find pages you want. Use "ctrl+f" to find keywords.
- Do not make judgement based on sources outside of Wikipedia or your prior knowledge. If the information is not in Wikipedia, click *Not-supported*.
- For the "Verify & Edit" step, do not add a new information unless necessary. For instance, if the bio says Bob attended Yale University whereas in fact Bob attended New York University, then you should fix it to Bob attended New York University. If the bio does not mention anything about the university attendance, then you shouldn't add Bob attended New York University.
- Subjective statements: If the information is reasonable based on the context in Wikipedia, then consider it as correct.
  - ex 1) His breakthrough came with the leading role in Parasite and you are not sure if Parasite is the *most* important breakthrough. → Consider it as *Supported* if Parasite is at least "one of" the important career of the person. However, if there are clearly other films/dramas that are more important than Parasite, usually according to the first paragraph of Wikipedia, you should mark it as *Not-supported* and edit the sentence to include more important movies.
  - ex 2) He is known for his friendly and down-to-earth personality. → Unless Wikipedia says something similar about personality, consider it as *Not-supported*.
- While we require to select only one supporting sentence, it is possible supporting the fact requires multiple sentences. We strongly recommend choosing one that is most important, even if not sufficiently supports the fact. However, there might be cases where it is impossible to do so. For instance, the fact He died at the age of 83 may be only possible to be verified by reading one sentence about the date of birth and the other sentence about the date of death. Then, either (a) select a section if there is a section that includes both information, or (b) select a sentence that is not already chosen for previous facts in the same sentence or for previous sentences (i.e., the sentence that is more exclusive).

Warnings

- Make sure to *accept* the HIT before doing anything! If you refresh before submitting the HIT, you will lose the progress! Make sure not to refresh before clicking the submit button.
- Once you accept the HIT, it will expire in 2 hours. Therefore, make sure to finish one HIT within 2 hours. If you want to reset the timer, you can "return" the HIT and grab it again. (However, don't do it if you already have made some pregress, since your progress won't be saved.)
- Wikipedia server is sometimes down. Please do not proceed and contact us in this case, and we will fix the server. Sorry about the inconvenience.
- There are sometimes issues with missing whitespaces or characters in Wikipedia. Please make the best judgement, keeping this in mind.

Figure 6: Instructions for data annotation in Section 4. We also provided a demonstration video, and gave feedback 1-1 during the qualification task.

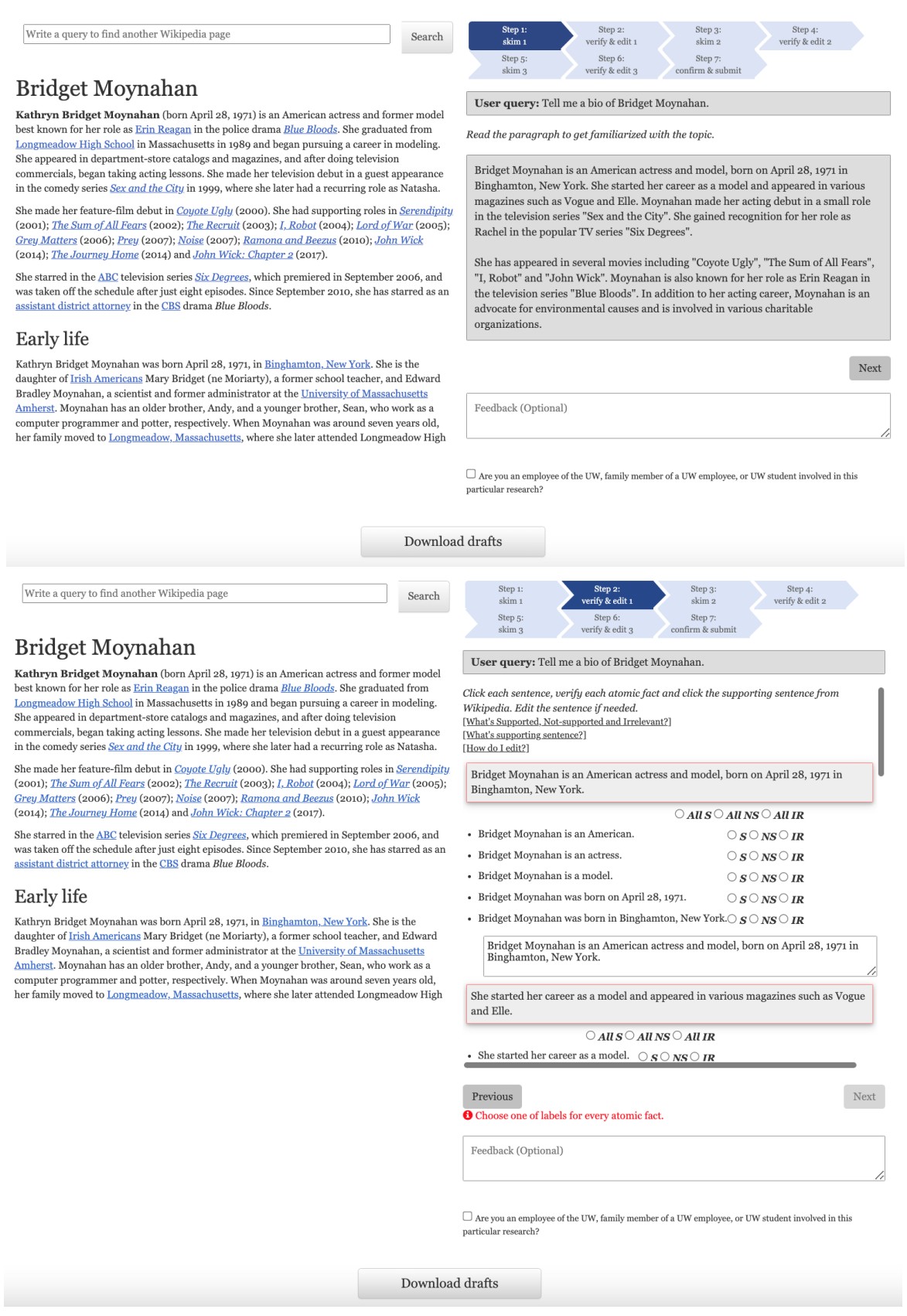

Figure 7: An interface for data annotation in Section 4. Annotators were able to navigate Wikipedia on the left. They annotate three pieces of generations from three LMs for the same prompt in one HIT since it saves time. Since completing one HIT takes considerable amount of time (25min), we added a function that allows saving their work at any stage in the middle of the HIT.