# OpenReview forum: "FActScore: Fine-grained Atomic Evaluation of Factual Precision in Long Form Text Generation"
_EMNLP/2023/Conference — EMNLP 2023 Main_

### Official Review · Reviewer_4um9 · 2023-07-30

**Soundness:** 4

**Excitement:**

3: Ambivalent: It has merits (e.g., it reports state-of-the-art results, the idea is nice), but there are key weaknesses (e.g., it describes incremental work), and it can significantly benefit from another round of revision. However, I won't object to accepting it if my co-reviewers champion it.

**Paper Topic And Main Contributions:**

This paper addresses the problem of evaluating the factuality of long-form text that is generated by large language models.

The authors propose FACTSCORE, a new scoring mechanism that takes into consideration each of atomic facts in the generated text.
FACTSCORE enables more fine-grained evaluation and comparison of the models.

Since a straightforward way of annotating data by human annotators is time-consuming and costly, the authors also propose an automatic approach of approximately estimating the value of FACTSCORE by using other models for retrieval and prediction of whether or not each atomic fact is supported by the pre-defined knowledge source.

Thorough analysis of the estimation for 12 different LMs indicates some interesting findings on correlation between factual accuracy and the model size, training data and instruction tuning, etc.



**Questions For The Authors:**

- In Page 3, line 484, "and ChatGPT for PerplexityAI."  In Table 3, LLAMA is the best for PerplexityAI with ER=0.1, so my understanding is that the authors choose ChatGPT (with ER=0.8) because it preserves the ranking while LLAMA does not. Is it correct?

- I wonder if the model size is dominant in factual precision because ChatGPT (GPT-3.5 or InstructGPT) and GPT-4 are known to be much larger than the other models discussed in the paper though no exact information on model sizes is available for them. Could you elaborate more about this if you hit upon anything? I read through the paper including Appendix and could find relevant discussions only in 4.3.2, but may have missed something important about the model size.


**Reasons To Accept:**

- FACTSCORE is a novel formulation of evaluation of factuality and addresses the problem of existing sentence-level evaluation methods.
- Though most of the details are in the Appendix, the paper is very well organized and each section is clearly explained.


**Reasons To Reject:**

- It is not explicitly stated how the findings discussed in Section 4 are utilized for deeper understanding of a model for factuality.
- Since analysis was done over only People biographies and Wikipedia for prompts and a knowledge source respectively, it is not clear how universal the findings are, which is also pointed out by the authors.


**Reproducibility:**

3: Could reproduce the results with some difficulty. The settings of parameters are underspecified or subjectively determined; the training/evaluation data are not widely available.

**Reviewer Confidence:**

3: Pretty sure, but there's a chance I missed something. Although I have a good feel for this area in general, I did not carefully check the paper's details, e.g., the math, experimental design, or novelty.

**Typos Grammar Style And Presentation Improvements:**

- Page 3, line 217: I think that [a is supported by C] can be written by the indicator function https://en.wikipedia.org/wiki/Indicator_function  . Or simply f(y) can be written as |{a \in A_y | a is supported by C}| / |A_y| .
- Page 6, line 406: across five variants -> "four" variants?

---

> ### Author Rebuttal · Authors · 2023-08-29
>
> Thank you for your insightful feedback and comments. We are glad you found FActScore  a novel formulation of evaluation and the paper well-written.
>
> > It is not explicitly stated how the findings discussed in Section 4 are utilized for deeper understanding of a model for factuality.
>
> We included in-depth discussion with respect to human evaluation (in Section 3.4) and a large set of LMs (in Section 4.3). If the reviewer has suggestions on more discussion, we are happy to incorporate it in the next version of the paper.
>
> > Since analysis was done over only People biographies and Wikipedia for prompts and a knowledge source respectively, it is not clear how universal the findings are, which is also pointed out by the authors.
>
> FactScore on its own is not limited to people's biographies — our main contribution is a general evaluation scheme which involves atomic fact decomposition, and evaluating the factuality of individual atomic facts. While in our paper we focus on biographies, as a proof of concept we ran some additional experiments in the “NLP domain” to show its generalizability.
>
> Our procedure was the following: (1) we manually wrote 10 prompts asking about NLP papers: “Tell me a summary of <paper_title>”; (2) we obtained responses from ChatGPT; (3) we next ran FActScore against an ACL anthology as a knowledge source, and (4) finally, we computed an error rate (ER)—a difference between humans’ validation (labeled by authors) and the model’s validation—as we did in Section 4. The ER is 7.41 (FactScore from humans: 66.20; FactScore from the model: 73.61), which is comparable to ER values in people bios shown in Table 3. This strongly suggests that FactScore can generalize beyond people's biographies.
>
> Overall, more thorough analysis in non-biography domains is an interesting direction for future work since it's possible that different models will be factual to different extents on different domains.
>
> > In Page 3, line 484, "and ChatGPT for PerplexityAI." In Table 3, LLAMA is the best for PerplexityAI with ER=0.1, so my understanding is that the authors choose ChatGPT (with ER=0.8) because it preserves the ranking while LLAMA does not. Is it correct?
>
> This is correct, since ER 0.8 is still considered to be very low. Generally, our recommendation is to use a few different variants of the estimator to make sure they give consistent results, which we verify to be true in Section 4.3.
>
> > I wonder if the model size is dominant in factual precision because ChatGPT (GPT-3.5 or InstructGPT) and GPT-4 are known to be much larger than the other models discussed in the paper though no exact information on model sizes is available for them. Could you elaborate more about this if you hit upon anything? I read through the paper including Appendix and could find relevant discussions only in 4.3.2, but may have missed something important about the model size.
>
> This is a great point. We do believe it is likely to be the case that some models are more factual than others because they are larger, which is in agreement with prior work that shows that larger LMs memorize factual knowledge better (CITE). We did not make this comment about OpenAI models since their sizes are unknown, and only made a relevant comment for open-sourced models whose sizes are known (Section 4.3.2).
>
> > Typos Grammar Style And Presentation Improvements
>
> Thank you for pointing out typos and comments on presentations. They will be incorporated in the next version of the paper.

---

### Official Review · Reviewer_Bgsm · 2023-08-03

**Soundness:** 4

**Excitement:**

4: Strong: This paper deepens the understanding of some phenomenon or lowers the barriers to an existing research direction.

**Paper Topic And Main Contributions:**

This paper proposes an evaluation method of factual precision called FACTSCORE, the percentage of pieces of information as atomic facts supported by a given knowledge source.  Since it is expensive to calculate this evaluation metric manually, this paper also proposes a method for automatic evaluation.

The evaluations of several commercial language models showed that
  1. infrequent entities tend to have lower  FACTSCORE,
  1. FACTSCORE of facts mentioned later in generated text tends to be lower, and
  1. even PerplexityAI, which uses search results, has a low FACTSCORE.

The experiment results also showed that the proposed automatic evaluation method can estimate FACTSCORE with a relatively low error rate for human evaluation.
In addition, the automatic evaluation of FACTSCORE for 12 large language models showed several interesting tendencies such that
  1.  none of the models is very high,
  1.  GPT-4 and ChatGPT are comparable, and
  1. GPT-4 and ChatGPT are better than other published models.

**Questions For The Authors:**

- A: What kind and size of the corpus were the NP models trained?

- B: Is there any reason there are no NP results for ChatGPT in Table 3?

- C: The reviewer thinks it is difficult for LLMs to generate facts without referring to any sources.
Considering that, what do the authors think is more valuable to evaluate it in a setting that generates text including facts from given relevant sources?
  - This setting would also have the advantage of evaluating "factual recall" from the facts mentioned in given relevant sources.

**Reasons To Accept:**

- This paper proposes a new score for evaluating factual precision and a method for automatically estimating the score without manual labor.
- The paper reports some interesting results by evaluating the scores of various large language models.
- There are plans to release it as open source.

**Reasons To Reject:**

- The generality needs to be clarified because the experiment was conducted only on Biographies.
- The method's validity needs to be clarified because the error compared to the case where the correct facts are considered, which is not written in knowledge sources, has not been evaluated.
- Since it is difficult for LLMs to generate facts without referring to any sources in principle, evaluating it in a setting that generates text including facts from given relevant sources may be more valuable.

**Reproducibility:**

3: Could reproduce the results with some difficulty. The settings of parameters are underspecified or subjectively determined; the training/evaluation data are not widely available.

**Reviewer Confidence:**

3: Pretty sure, but there's a chance I missed something. Although I have a good feel for this area in general, I did not carefully check the paper's details, e.g., the math, experimental design, or novelty.

---

> ### Author Rebuttal · Authors · 2023-08-29
>
> Thank you for your detailed comments and feedback. We are glad you found that the results and analysis in the paper are interesting and that the evaluation metric is easy to use and open-source is a valuable contribution.
>
> > The generality needs to be clarified because the experiment was conducted only on Biographies.
>
> FactScore on its own is not limited to people's biographies — our main contribution is a general evaluation scheme which involves atomic fact decomposition, and evaluating the factuality of individual atomic facts. While in our paper we focus on biographies, as a proof of concept we ran some additional experiments in the “NLP domain” to show its generalizability.
>
> Our procedure was the following: (1) we manually wrote 10 prompts asking about NLP papers: “Tell me a summary of <paper_title>”; (2) we obtained responses from ChatGPT; (3) we next ran FActScore against an ACL anthology as a knowledge source, and (4) finally, we computed an error rate (ER)—a difference between humans’ validation (labeled by authors) and the model’s validation—as we did in Section 4. The ER is 7.41 (FactScore from humans: 66.20; FactScore from the model: 73.61), which is comparable to ER values in people bios shown in Table 3. This strongly suggests that FactScore can generalize beyond people's biographies.
>
> Overall, more thorough analysis in non-biography domains is an interesting direction for future work since it's possible that different models will be factual to different extents on different domains.
>
> > The method's validity needs to be clarified because the error compared to the case where the correct facts are considered, which is not written in knowledge sources, has not been evaluated.
>
> To quantify how big of an issue this is, we ran new, small analysis experiments as follows. We randomly sampled 30 unsupported facts from ChatGPT on people whose categories are either “rare” or “very rare” (based on Reviewer 4fZt’s point that this would be most problematic for rare entities), and then validated against the entire web. We found 10% (3 out of 30 facts) are true even if not presented in Wikipedia. As statistics indicate, these cases are relatively rare and have negligible impact to the FactScores returned by our model.
>
> We also highlight that we found that Wikipedia has a high coverage and mentions most of the important information that we were able to find from any other sources on the web. This is in agreement with prior work that treated Wikipedia as a general knowledge source under the same reason [1, 2, and more].
>
> Please also note that if a fact is not mentioned anywhere on the web, there would be no way one can validate it (which is why we use the term “supported” and “unsupported” rather than “true” and “false”), and ChatGPT would not be able to generate it either.
>
> [1] Chen et al. ACL 2017. Reading Wikipedia to Answer Open-Domain Questions.
> [2] Petroni et al. NAACL 2021. KILT: a Benchmark for Knowledge Intensive Language Tasks.
>
> > Since it is difficult for LLMs to generate facts without referring to any sources in principle, evaluating it in a setting that generates text including facts from given relevant sources may be more valuable.
>
> We evaluated models using the Wikipedia article in context in our Perplexity AI setting, which is one of the three models we evaluated. Perplexity AI is a retrieval augmented system that searches for relevant reference documents from the English Wikipedia, feeds them together with the original prompt to ChatGPT, and generates a response [1]. Indeed, Perplexity AI has a much better FActScore than ChatGPT (71.5 vs 58.3).
>
> [1] - A podcast with Perplexity AI’s CEO confirming this: https://youtu.be/YSYQZZu4MEM?si=GTK1yZO52348BnOL&t=323
>
> > A: What kind and size of the corpus were the NP models trained?
>
> The NP model is trained on Wikipedia and news, on top of RoBERTa, which is trained on Wikipedia and books.
>
> > B: Is there any reason there are no NP results for ChatGPT in Table 3?
>
> In fact, the NP-only model does not use any language model, although we list it “I-LLAMA” in Table 3. We will modify the table to prevent the confusion.
>
> > C: The reviewer thinks it is difficult for LLMs to generate facts without referring to any sources. Considering that, what do the authors think is more valuable to evaluate it in a setting that generates text including facts from given relevant sources? This setting would also have the advantage of evaluating "factual recall" from the facts mentioned in given relevant sources.
>
> We agree with the reviewer that it is valuable to evaluate LLMs with relevant sources in context. However, we believe it is equally important to evaluate models like ChatGPT *without* the Wikipedia article included in context. By default, ChatGPT does not assume access to the internet, and several users solely rely on its parametric knowledge for information-seeking needs. Overall, we believe it is critical for a chatbot to output factually correct statements, or acknowledge that it is not familiar with the information (rather than hallucinate).
>
> > Reproducibility: 2
>
> We include all details to reproduce results in the Appendix, and will open-source the code and the data through the PIP package, as highlighted in the abstract.

---

### Official Review · Reviewer_4fZt · 2023-08-04

**Soundness:** 4

**Excitement:**

4: Strong: This paper deepens the understanding of some phenomenon or lowers the barriers to an existing research direction.

**Paper Topic And Main Contributions:**

The paper investigates hallucinations in generated biographies. The main contribution is in the in depth analysis of hallucinations in the generated biographies, with the added benefit of considering SOTA language models. FACTScore, the metric proposed by the authors, is simply the average truth score of a sentence composed of several facts, where each fact gets 0 (not true) or 1 (true) score. This score is computed by humans in a large scale evaluation, but the authors also propose a few models that can achieve good results.

**Questions For The Authors:**

A. In which measure Figure 2 (rare entities has less correct biography) could be explained through the fact that the human annotators did not find those particularly facts on the Wikipedia page? Did you look if there is a correlation between the type of unsupported fact (page level contradiction or annotation level) and the popularity/completeness of the page of entities? Is it possible that for the rare entities the fact is true, just not present on Wikipedia?

B. Did you consider the setting in which the LM receives in the prompt the Wikipedia page of the entity and has the task of creating a biography? The model is very likely to produce more accurate results, which will be consistent with the findings you have when you create the automatic measure.

**Reasons To Accept:**

I found the paper well written and easy to follow. In my opinion, the work is sound and the experimental evaluation rigorous. In addition, the problem studied is interesting. The authors invested a lot of money and effort in the experimental evaluation of hallucinations, which is the main strength.  In particular, they do a human evaluation of biography generation for three models: InstructGPT, ChatGPT, PerplexityAI on 183 generations (biographies) from each model. The humans split the generations in smaller units of content that can be easily verified by checking the Wikipedia page of the person.

**Reasons To Reject:**

The FactScore metric is a very simple metric that cannot be counted as a contribution, as it hides a complex discussion of what constitutes a fact and when a fact is truly supported by an underlying KB/document. For example, if a sentences is not correctly split in the underlying parts, evaluating its truth value is non trivial, for example: President Trump no longer lives in the White House. Here we have 2 facts (Trump, is, President), (Trump, no longer lives in, White House). Also negations are an added layer of difficulty as often KBs or documents do not have all possible negations if any. Hence, the true contribution of the paper is more in the human analysis of the hallucinations of current models.
The human analysis of hallucinations has also the drawback that it does not considers that some facts might be true, but not present in the Wikipedia page. This can be more serious for entities with shorter Wikipedia pages, for which the authors find that models tend to produce less accurate results.
Finally, models might produce much more accurate results if given in input the Wikipedia page - an hypothesis which is very interesting to test.

**Reproducibility:**

4: Could mostly reproduce the results, but there may be some variation because of sample variance or minor variations in their interpretation of the protocol or method.

**Reviewer Confidence:**

4: Quite sure. I tried to check the important points carefully. It's unlikely, though conceivable, that I missed something that should affect my ratings.

---

> ### Author Rebuttal · Authors · 2023-08-29
>
> Thank you for your thorough comments and feedback. We are glad you found that the problem tackled in the paper is interesting, the idea of using smaller units is valuable, and the work has rigorous experiments.
>
> > The FactScore metric is a very simple metric that cannot be counted as a contribution.
>
> We believe introducing a simple and easy-to-use metric is a valuable and important contribution to the community that fosters reproducible and scalable evaluation, as it has been the case in past work like BERTScore [1]. Moreover, we think breaking down into atomic facts and validating each is not a trivial idea and has not been done in prior work, e.g., prior work considers a coarser-level validation, as described in Section 2.
>
> [1] Zhang et al. BERTScore: Evaluating Text Generation with BERT. ICLR 2020.
>
> > It hides a complex discussion of what constitutes a fact and when a fact is truly supported by an underlying KB/document.
>
> We agree with the reviewer that what constitutes a fact and whether it's truly supported by an underlying document are indeed non-trivial questions. We included such discussion in the Limitation section, and highlight our contribution is to introduce an evaluation scheme that (a) works well in the domain where what constitutes “supported” is relatively straight-forward, and (b) is easily extendable.
>
> > For example, if a sentence is not correctly split in the underlying parts, evaluating its truth value is non trivial, for example: President Trump no longer lives in the White House. Here we have 2 facts (Trump, is, President), (Trump, no longer lives in, White House).
>
> We agree that decomposition of an input with a highly complex linguistic structure is generally harder. However, in practice, model generations tend to have much simpler structure so such cases are very rare. For instance, in our experiments, only 3 out of 1241 sentences from ChatGPT include a negation.
>
> Moreover, there is no assumption we are making that would prevent the model from not being able to handle negations. In fact, we found that our atomic fact decomposition model can handle them. See [the screenshot](https://ibb.co/Fb35259) on how our model correctly handles three sentences from ChatGPT that include a negation.
>
> We will include more in-depth discussion on this matter in the Limitation section in the next version of the paper.
>
> > Also negations are an added layer of difficulty as often KBs or documents do not have all possible negations if any.
>
> Since we are using a language model to validate each fact, and the language model can generalize reasonably well, it can make a correct validation even if the knowledge source does not include an explicit negation.
>
> > The human analysis of hallucinations has also the drawback that it does not consider that some facts might be true, but not present in the Wikipedia page. This can be more serious for entities with shorter Wikipedia pages, for which the authors find that models tend to produce less accurate results.
>
> To quantify how big of an issue this is, we ran new, small analysis experiments as follows. We randomly sampled 30 unsupported facts from ChatGPT on people whose categories are either “rare” or “very rare”, and then validated against the entire web. We found 10% (3 out of 30 facts) are supported even though they are not supported in Wikipedia. An example is “[Hibo] Wardere published her memoir titled <Cut: One Woman's Fight Against FGM in Britain Today>" which is not mentioned in Wikipedia but is found from Google Books. Since these cases are rare, we believe this does not change the conclusion from the paper.
>
> Nonetheless, we found that Wikipedia has a high coverage and mentions most of the important information that we were able to find from any other sources on the web. This is in agreement with prior work that treated Wikipedia as a general knowledge source under the same reason [1, 2, and more].
>
> [1] Chen et al. ACL 2017. Reading Wikipedia to Answer Open-Domain Questions.
> [2] Petroni et al. NAACL 2021. KILT: a Benchmark for Knowledge Intensive Language Tasks.
>
> > A. In which measure Figure 2 (rare entities has less correct biography) could be explained through the fact that the human annotators did not find those particularly facts on the Wikipedia page? Did you look if there is a correlation between the type of unsupported fact (page level contradiction or annotation level) and the popularity/completeness of the page of entities? Is it possible that for the rare entities the fact is true, just not present on Wikipedia?
>
> As mentioned earlier, based on our analysis on “rare” and “very rare” categories (we conducted this analysis for “rare” and “very rare” because we agree with the intuition that the completeness of Wikipedia will be better in other categories), only 10% were the cases where the fact can be supported by other sources on the web but are not supported in Wikipedia due to its incompleteness. This does not change the conclusion from the paper, e.g., adding 10% to the “Rare” and “Very rare” in Figure 2 does not change the overall trend of the figure.
>
> > Finally, models might produce much more accurate results if given in input the Wikipedia page - an hypothesis which is very interesting to test.
>
> > B. Did you consider the setting in which the LM receives in the prompt the Wikipedia page of the entity and has the task of creating a biography? The model is very likely to produce more accurate results, which will be consistent with the findings you have when you create the automatic measure.
>
> We evaluated models using the Wikipedia article in the input context in our Perplexity AI setting, which is one of the three models evaluated in Section 3. Perplexity AI is a retrieval augmented system that searches for relevant reference documents from the English Wikipedia, feeds them together with the original prompt to ChatGPT, and generates a response [1]. Indeed, Perplexity AI has a much better FActScore than ChatGPT (71.5 vs 58.3). Overall, we think it is important to evaluate settings both with and without the relevant sources included in context, especially given that no Internet access is the default setting of many LM applications like ChatGPT.
>
> [1] - A podcast with Perplexity AI’s CEO confirming this: https://youtu.be/YSYQZZu4MEM?si=GTK1yZO52348BnOL&t=323

---

### Meta-Review · Area_Chair_mVgK · 2023-09-11

**Recommendation:** 4

**Metareview:**

Pros:
1. Paper is well written and easy to follow. Work is sound and the experimental evaluation rigorous.
2. Problem is interesting and critical in the area of text-based generative AI -- a new score for evaluating factual precision  of outputs from LLMs automatically.
3. Simple metric. Insights from evaluation of FACTSCORE for 12 large language models are interesting.
4. Contribution to open source is a strong positive.

Cons:
1. Work is very empirical, sometimes leading to lack of concrete definitions, e.g., "facts". But this is rightly pointed out in Limitations section. And EMNLP is an empirical venue.
2. Generalization is a problem since the work deals with 1 domain and 1 knowledge base.
3. Incompleness of the knowledge base could hurt the evaluation. But the work stands on its own assuming that incompleteness of knowledge base is a separate problem to solve.

Suggestions:
For the last 2 points in cons section: Experiments done as part of rebuttal are not very rigorous and done on small scale. While authors are encouraged to include them as part of appendix in the revised submission, this weakness should be explicitly called out in the Limitations section.

---

### Decision · Program_Chairs · 2023-10-07

**Decision:**

Accept-Main

**Comment:**

Pros:
1. Paper is well written and easy to follow. Work is sound and the experimental evaluation rigorous.
2. Problem is interesting and critical in the area of text-based generative AI -- a new score for evaluating factual precision  of outputs from LLMs automatically.
3. Simple metric. Insights from evaluation of FACTSCORE for 12 large language models are interesting.
4. Contribution to open source is a strong positive.

Cons:
1. Work is very empirical, sometimes leading to lack of concrete definitions, e.g., "facts". But this is rightly pointed out in Limitations section. And EMNLP is an empirical venue.
2. Generalization is a problem since the work deals with 1 domain and 1 knowledge base.
3. Incompleness of the knowledge base could hurt the evaluation. But the work stands on its own assuming that incompleteness of knowledge base is a separate problem to solve.

Suggestions:
For the last 2 points in cons section: Experiments done as part of rebuttal are not very rigorous and done on small scale. While authors are encouraged to include them as part of appendix in the revised submission, this weakness should be explicitly called out in the Limitations section.